Corrected: Author correction

# A comprehensive model for assessment of liver stage therapies targeting *Plasmodium vivax* and *Plasmodium falciparum*

Alison Roth [1], Steven P. Maher [1,2], Amy J. Conway[1], Ratawan Ubalee[3], Victor Chaumeau[4,5], Chiara Andolina[4,5], Stephen A. Kaba[6], Amélie Vantaux[7], Malina A. Bakowski[8], Richard Thomson-Luque[1], Swamy Rakesh Adapa[1], Naresh Singh[1], Samantha J. Barnes[1], Caitlin A. Cooper[2], Mélanie Rouillier[9], Case W. McNamara[8], Sebastian A. Mikolajczak[10], Noah Sather[10], Benoît Witkowski[8], Brice Campo[9], Stefan H.I. Kappe[10], David E. Lanar[6], François Nosten [4,5], Silas Davidson[3], Rays H.Y. Jiang [1], Dennis E. Kyle [1,2] & John H. Adams [1]

Malaria liver stages represent an ideal therapeutic target with a bottleneck in parasite load and reduced clinical symptoms; however, current in vitro pre-erythrocytic (PE) models for *Plasmodium vivax* and *P. falciparum* lack the efficiency necessary for rapid identification and effective evaluation of new vaccines and drugs, especially targeting late liver-stage development and hypnozoites. Herein we report the development of a 384-well plate culture system using commercially available materials, including cryopreserved primary human hepatocytes. Hepatocyte physiology is maintained for at least 30 days and supports development of *P. vivax* hypnozoites and complete maturation of *P. vivax* and *P. falciparum* schizonts. Our multimodal analysis in antimalarial therapeutic research identifies important PE inhibition mechanisms: immune antibodies against sporozoite surface proteins functionally inhibit liver stage development and ion homeostasis is essential for schizont and hypnozoite viability. This model can be implemented in laboratories in disease-endemic areas to accelerate vaccine and drug discovery research.

[1] Department of Global Health, College of Public Health, Center for Global Health and Infectious Diseases Research, University of South Florida, 3720 Spectrum Blvd 404, Tampa, FL 33612, USA. [2] Center for Tropical and Emerging Global Diseases, University of Georgia, 500 DW Brooks Dr. Suite 370, Athens, GA 30602, USA. [3] Department of Entomology, Armed Forces Research Institute of Medical Sciences (AFRIMS), 315/6 Rajvithi Rd, Bangkok 10400, Thailand. [4] Centre for Tropical Medicine and Global Health, Nuffield Department of Medicine, University of Oxford, Oxford, UK. [5] Shoklo Malaria Research Unit, Mahidol Oxford Research Unit, Faculty of Tropical Medicine, Mahidol University, 68/30 Bantung Rd, Mae Sot, Tak 63110, Thailand. [6] Malaria Vaccine Branch, Walter Reed Army Institute of Research, 503 Robert Grant Ave, Silver Spring, MD 20910, USA. [7] Malaria Molecular Epidemiology Unit, Institut Pasteur du Cambodge, 5 Boulevard Monivong-PO Box 983, Phnom Penh 12 201, Cambodia. [8] California Institute for Biomedical Research (Calibr), 11119N. Torrey Pines Rd, Suite 100, La Jolla, CA 92037, USA. [9] Medicines for Malaria Venture, Pré-Bois Rd 20, Meyrin 1215, Switzerland. [10] Center for Infectious Disease Research, 307 Westlake Ave N Suite 500, Seattle, WA 98109, USA. These authors contributed equally: Alison Roth, Steven P. Maher. Correspondence and requests for materials should be addressed to J.H.A. (email: jadams3@health.usf.edu)

Malaria is a major global disease with almost half of the world's population at risk, resulting in an estimated 216 million cases and 445,000 deaths in 2016[1]. The two most prevalent causes of malaria are apicomplexan parasites *P. falciparum*, the most virulent and dominant species in Sub-Saharan Africa, and *P. vivax*, with the widest geographical distribution and major economic impact[2–4]. *Plasmodium* sporozoites initiate infections when injected into the dermis by a female anopheline mosquito and then use a molecular motor-driven mechanism to rapidly enter the human circulatory system and translocate across the liver sinusoid[5–7]. After invasion of hepatocytes, liver-resident parasites undergo asexual schizogony to form tens of thousands of merozoites. Following merozoite egress from the infected liver cell, the parasites escape into the hepatic vein to infect erythrocytes where they asexually replicate in circulation, leading to geometric population expansion and the clinical symptoms of malaria. Although it is the blood-stage infection that causes clinical disease, the sporozoite and the liver stage (LS) forms, which together are referred to as pre-erythrocytic (PE) stages, represent a vulnerable bottleneck for therapeutic interventions to prevent malaria[8,9]. Therefore, chemotherapeutic and immunoprophylactic interventions have converged on targeting *P. vivax* and *P. falciparum* PE stages as a strategy to block progression to clinical malaria and transmission[10].

The biology of the *P. vivax* and *P. falciparum* LS forms fundamentally differ as some *P. vivax* parasites will remain quiescent as uninucleate stages termed hypnozoites[11]. Hypnozoites are not susceptible to the mechanism of action of most antimalarial drugs and can persist for weeks, months, or even years before an unknown re-activation mechanism stimulates completion of development and a symptomatic blood-stage infection[12]. In addition, *P. vivax* is able to rapidly form transmissible gametocytes in circulation before presentation of clinical symptoms[13]. Therefore, an effective malaria elimination toolbox has been proposed consisting of a multi-stage drug with hypnozonticidal activity and a highly efficacious vaccine conferring life-long sterile immunity; however, neither of these tools is currently available[10]. Malaria control is now focused on treating the symptomatic blood and transmission stages with front-line drugs of Artemisinin Combination Therapies (ACTs) for falciparum malaria and predominately chloroquine (CQ) for vivax malaria[1,14]. Prophylactic regimens of atovaquone and proguanil (Malarone®) target only the LS schizont, while the only chemotherapeutic intervention currently capable of targeting hypnozoites are 8-aminoquinolines, such as primaquine and tafenoquine[15,16]. Unfortunately, use of 8-aminoquinolones is contraindicated in many malaria endemic countries because of its toxicity in individuals with some glucose-6-phosphate dehydrogenase (G6PD) polymorphisms, restricting mass drug administration campaigns in regions where high-risk favisms are common[17]. In regards to malaria vaccinology, development of PE vaccines has focused on the initial stages of infection targeting antibodies to the sporozoite surface to neutralize parasite migration to the liver and consequently the disease-causing blood stage[18–20]. However, vaccines to prevent malaria have lagged far behind drug development efforts as only one vaccine for *P. falciparum* has been licensed, RTS,S-S/ASO1 or Mosquirix™[21]. In Phase III clinical field trials Mosquirix™ showed a temporary, age-specific response with only partial protection[22]. Meanwhile, vaccines for *P. vivax* remain mostly in the pre-clinical discovery phase of development and only a few candidates have progressed into initial clinical trials[23].

In vitro PE assays are essential for preclinical assessment of novel vaccines and drugs, yet currently available PE assays are inadequate for meeting the demands of a genuine PE screening effort[24]. Historically, many studies of *Plasmodium* liver models used human hepatoma lines, which are deficient in specific surface receptors present on primary human hepatocytes (PHHs) that are required for *Plasmodium* sporozoite invasion, resulting in poor invasion rates[25,26]. Furthermore, LS formation within hepatoma cells is atypical compared to that noted in animal models as the schizants are smaller and cannot be as easily distinguished from hypnozoites and persistent proliferation of these host cells hinders image-based analysis[27]. More recent studies have used fresh, cryopreserved PHHs, or human iPSC-derived hepatocyte-like cells in a 96-well plate co-culture model, yet LS development rates (LS parasites per sporozoite inoculum) remained low despite sporozoite infection loads 10-fold higher than what we report herein[25,28–30]. Animal models engrafted with PHHs offer excellent *Plasmodium* LS development but intrinsically high costs and low-throughput hinders the use of this model for drug discovery[31,32].

In this report we describe a robust anti-PE therapeutic screen streamlined for *P. vivax* and *P. falciparum* using a PHH culture system comprised entirely of commercially available 384-well plates and cell culture reagents. Reducing to a 384-well microtiter format promotes key morphological and functional characteristics of native in situ hepatocytes and allows for high-resolution imaging, seamless image acquisition with faster imaging speed, and integration of automated high-content image analysis. Identification of optimal sporozoite isolation and hepatocyte culture methods resulted in highly reproducible formation of hundreds of LS parasites in each well of a 384-well plate, making the model ideal for *P. vivax* and *P. falciparum* PE bioassays. Implementation of our PHH culture system in laboratories located in and outside of malaria endemic areas demonstrates this novel PE model will help fill a critical technology gap hindering advancement of PE-active therapies and lay the foundation for next-generation malaria control and elimination.

## Results

**Functional human hepatocytes in a commercial 384-well plate.**
We discovered that the small-scale collagen-treated surface area of particular commercially available 384-well plates coupled with our methodology provides a suitable microphysiological environment for long-term cultivation of PHHs (Fig. 1a, Supplementary Fig. 1a). The PHH donor lots PDC and NLX, used for the majority of studies described herein, were seeded at a density of $1.8 \times 10^4$ live cells per well to achieve attachment of $\sim 1 \times 10^4$ cells per well. High-content imaging (HCI) of the cultured PHHs revealed individual cells in a confluent layer organized into lobule-like multicellular units. A rapid re-acquisition of primary cell characteristics occurred within two days, including visible transport of biliary metabolites and active mitochondria, and then remained stable for at least 30 days (Fig. 2a).

Functionality and metabolic activity of cultured PHHs were further characterized by several biometrics. Albumin production leveled at day 4 and remained stable thereafter with slight variation between 2 and $4 \, \mathrm{ng \, ml^{-1}}$ for a duration of 30 days (Fig. 2b, Supplementary Fig. 1b). Furthermore, PHHs maintained a CYP3A4 response to Rifampicin between days 4 and 20 along with stable factor IX expression for 3 weeks (Supplementary Fig. 1c, d). Hepatobiliary transport (active and inactive) of the bile canaliculi was measured by staining with CellTracker™ Green CMFDA and quantified by image analysis. This biomarker passes into hepatocytes whereupon it is trapped as cell-impermeable glutathione methylfluorescein (GS-MF) that is subsequently actively transported into the bile canaliculi. Accumulation of this biliary metabolite can be used to quantify the functional

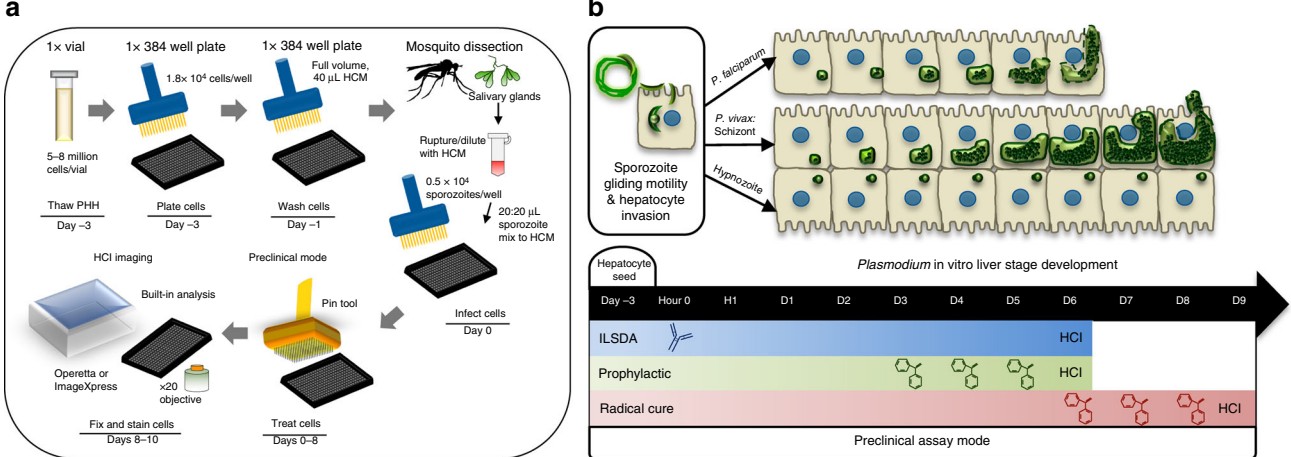

**Fig. 1** Experimental workflow for multispecies drug and vaccine preclinical assays. **a** A single vial of a pre-established cryopreserved primary human hepatocyte (PHH) donor is sufficient to seed a full 384-well plate and run 360 single-point therapeutic candidates. By day 3 post-seed, PHHs show in vivo-like phenotypes and are suitable for *Plasmodium* sporozoite infection. Defined preclinical assay modes are standardized using semi-automated hand pipettes and a 384-well pin tool. Upon experiment fixation, plates are fluorescently stain with a specific LS biomarker and imaged on a high content imaging (HCI) system. **b** *Plasmodium* sporozoites exhibit gliding motility while traveling from the injection site in the dermis to liver. Contact with liver components, including Kupffer cells, highly sulfated heparan sulfate proteoglycans, and liver endothelium triggers proteolytic processing of surface proteins (i.e., circumsporozoite protein) and activation of hepatocyte invasion machinery (i.e., TRAP, TREP, P36). Antibody inhibition of invasion pathways can be assessed by pre-incubation of infectious sporozoites with test serum or purified antibody in vitro prior to infection of plated hepatocytes (ILSDA). Following successful invasion, including formation of a parasitophorous vacuole membrane, *P. vivax* differentiates into hypnozoites and liver schizonts, which complete development after day 9. Day 5 or 6 is an ideal endpoint for both *P. falciparum* and *P. vivax*, for ILSDA and prophylactic drug response assays, as *P. falciparum* completes development shortly thereafter, and *P. vivax* schizonts are then distinguishable from hypnozoites based on size and morphology. Radical cure drug response assays are ended at day 8 to allow for treatment and clearance of susceptible liver forms

activity of multi-drug resistance-associated protein 2 (MRP2) to serve as a surrogate indicator of hepatocyte polarization[28,33] (Supplementary Fig. 2). Characterization of transport by PHH lot PDC showed a steady increase of active bile canaliculi network formation within the monolayer of lobule-like microstructures coupled with continued growth in bile canaliculi length reaching maximum lengths of 122.90 μm with a mean length of 19.33 ± 9.01 μm (±standard deviation) on day 21 compared to a maximum length of 19.19 μm with a mean length of 11.90 ± 2.35 μm on day 2 ($n = 3$, $n' = 3$) (Fig. 2c, d). We assessed an additional biomarker of hepatocyte viability by characterizing the electrical potential between the interior and exterior of functioning mitochondria using cell-permeant dye tetramethylrhodamine (TMRM), which produces a bright signal within polarized mitochondria[34,35]. Measuring mean fluorescence intensity (MFI) of cytoplasmic stain and then comparing the daily time points indicated a 4-fold increase of mitochondrial activity from day 2 to 21 (Fig. 2e). Further image analysis revealed co-localization of high mitochondrial activity and bile canaliculi growth (Mander's coefficients M1, M2 = 1 and tM1, tM2 ≥ 0.3–0.8) with highest correlation during early bile duct formation on day 2 ($r = 0.72$) (Supplementary Fig. 3a–c)[36,37]. Altogether the in vivo-like physiological and morphological indicators of this in vitro liver platform were expected to be highly supportive for experimental studies of the major human malaria parasites, *P. vivax* and *P. falciparum*.

**Complete development of *P. vivax* and *P. falciparum* LS parasites.** To evaluate our model's suitability for experimental studies with human malaria parasites, we tested cryopreserved PHH donor lots from four companies and subjected each to an experimental matrix to include a range of seeding densities and sporozoite isolation methods. In our hands, three lots tested from Bioreclamation IVT (BIVT) supported higher *P. vivax* and

*P. falciparum* LS parasite development rates than lots tested from other sources (Supplementary Fig. 4a, Supplementary Table 1). We therefore decided to screen an additional 13 BIVT PHH donor lots for both *P. vivax* and *P. falciparum* development relative to lot PDC and the terminally-differentiated HepaRG hepatocyte line (as a negative control for LS development). Four donors, lots H, J, P, and Q, were significantly higher in LS development rate relative to PDC ($P < 0.0001$ and $P < 0.05$, two-way ANOVA followed by Dunnett's multiple comparisons) while two donors, lot F and N did not support *P. falciparum* or *P. vivax* LS parasite growth. We also discovered some donors, lots G, H, I, and P, appeared susceptible to only *P. falciparum* or *P. vivax* (Fig. 3a). Three PHH donors did not form a monolayer in the 384-well microplate.

Following characterization of *Plasmodium* spp. sporozoite infection rate, this expanded set of PHH donor lots were assessed for basal metabolic CYP activity (Supplementary Table 2, Supplementary Fig. 5) and hepatic functionality by measurement of bile canaliculi formation and mitochondrial activity over 14 days (Supplementary Table 3). Interestingly when measured over days 2–8, PHH donors found to have a stable mean TMRM intensity ( ≥ 3000 MFI), increase of cytoplasmic-stained cell area substantial active bile transport ( ≥ 30%), and a decrease of total hepatocytes appeared to support higher *Plasmodium* LS developmental rates. Furthermore, PHH donor lots with only a 2-fold increase in mean TMRM intensity and increased amount of inactive transport on day 8 ( ≥ 50%) had attenuated *Plasmodium* LS development with a reduction in mean LS schizont area. Lastly, the absence of LS parasite development was only observed in PHH donor lots with a decrease in active transport and minimal change in mean TMRM intensity from day 2 to 8 (Supplementary Table 3). After identification of suitable PHH donor lots, other microenvironmental factors that might influence sporozoite invasion and LS development were optimized. Hepatocyte seeding density and time between seed

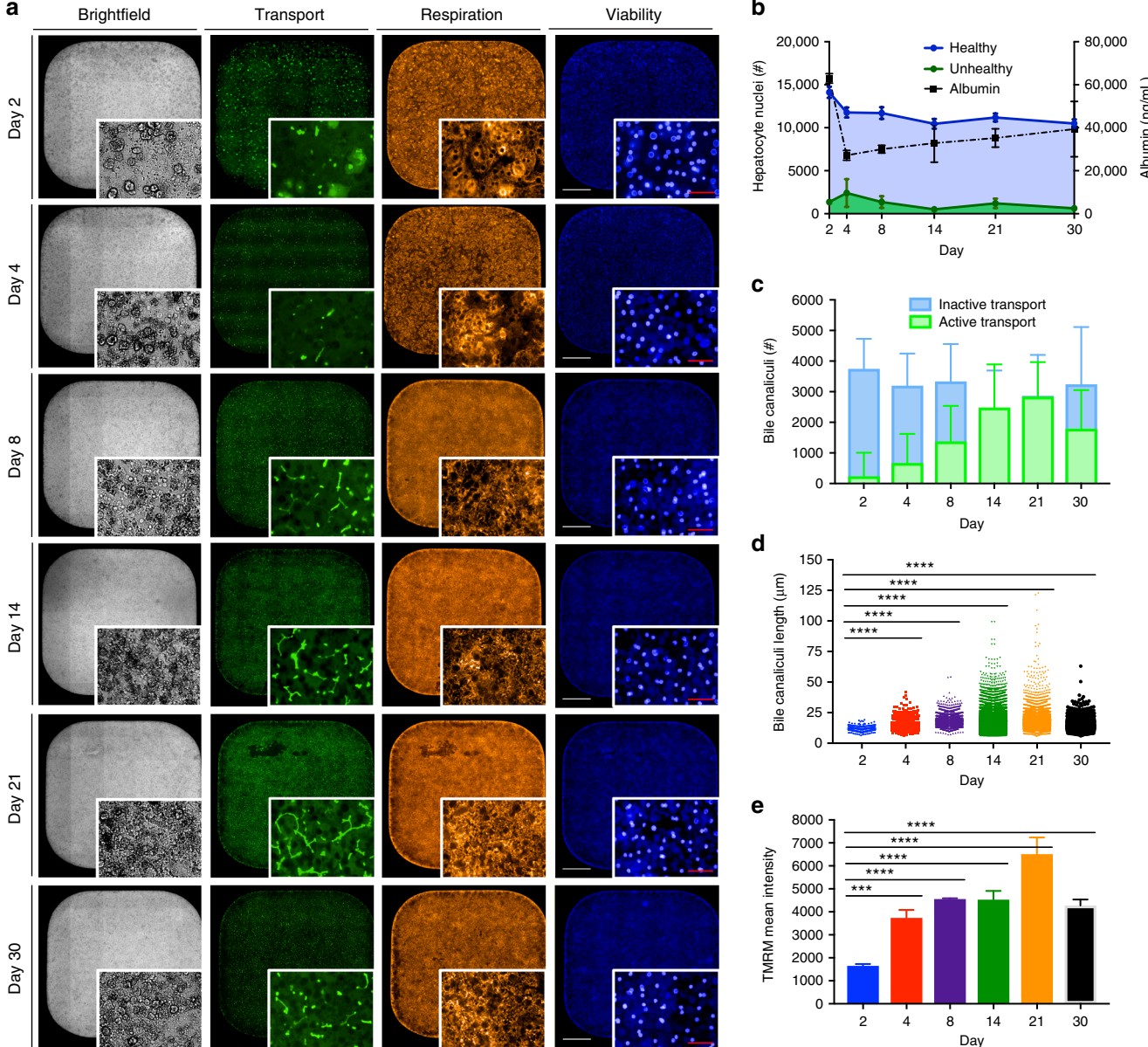

**Fig. 2** Functional assessment of cultured primary human hepatocytes (PHHs). **a** Photometric measurements of live cellular functions (transport, respiration, and viability) were imaged on days 2, 4, 8, 14, 21, 30 post seed revealing PHHs cultured in a 384-well plate remain stable for 30 days. **b** Condition of hepatocyte monolayer was determined by nuclei morphology, size, and fluorescence intensity, then classified as "healthy" and "unhealthy". A steady adhered monolayer of ~11,000 PHHs and high albumin expression (ng ml$^{-1}$) persisted for the duration of 30 days. **c, d** Image analysis combining mean intensity of CellTracker$^{TM}$ Green CMFDA with differential imaging filters allowed for bile canaliculi quantification, assignment of active vs. inactive transport, and identification of bile duct length (μm) where peak measurements were revealed on day 21 post seed. **e** The mean intensity of the fluorescent dye TMRM was used to measure mitochondrial activity with day 21 having the highest active mitochondrial sequestration. Graph bars represent means with s.d. for biological replicates ($n = 3$) and experimental replicates ($n = 6$, **b**, $n = 3$, **c, e**) or individual values were plotted (**d**). Statistical significance was calculated using an one-way ANOVA with Dunnett's multiple comparisons test to day 2 where statistical significance values are represented as $P < 0.0001$ (***) and $P < 0.0001$ (****). White scale bars represent 500 μm and red represents 50 μm

and sporozoite infection were identified as crucial variables; the highest *P. vivax* LS development rates resulted from a seeding density of $1.8 \times 10^4$ live PHHs yielding a monolayer of $~1.0 \times 10^4$ cells, followed by a delayed sporozoite infection until at least day 3 to allow for monolayer assembly (Supplementary Fig. 4b–d).

The *P. vivax* sporozoites produced from mosquitoes fed on clinically-isolated blood meals consistently yielded superior LS developmental rates (2–8.30%) compared to sporozoites from mosquito infections derived from in vitro-cultured *P. falciparum* NF54-WT (0.60–2%) and *P. falciparum* NF54-GFP (0.04–0.40%)

following an inoculum of $5.0 \times 10^3$ sporozoites (Fig. 3b, f, g). Characterization of day 6 LS parasites revealed synchronized growth of *P. falciparum* with 75.95% of the parasites having a mean parasite area of $73.82 \pm 6.52$ μm$^2$ ($n = 3$, $n' = 8$) (Fig. 3c, f). Inversely, *P. vivax* day 6 parasites developed asynchronously as the schizont population had larger, irregular parasite area compared to *P. falciparum* (Fig. 3c, g) ranging from $31.65 \pm 0.97$ to $1270 \pm 197.30$ μm$^2$ ($n = 3$, $n' = 26$). By day 8, some *P. falciparum* LS schizonts successfully completed maturation and released infectious merozoites leading to further variability in LS

quantification and size distribution (Supplementary Fig. 6). Furthermore, *P. vivax* LS schizonts continued to grow remarkably larger from day 6 and nearly doubled in area with 32.44% LS having a mean area of $2709 \pm 333.20\ \mu m^2$ ($n = 3$, $n' = 26$) (Fig. 3d, g and Supplementary Table 4). Importantly, from days 6 to 8,

*P. vivax* LS hypnozoites represented ~60% of the total LS parasites per well with a mean size of $34.55 \pm 2.37\ \mu m^2$ ($n = 3$, $n' = 26$) (Fig. 3e).

Developing LS forms of *P. vivax* and *P. falciparum* were analyzed by HCI using a panel of *Plasmodium* antibodies

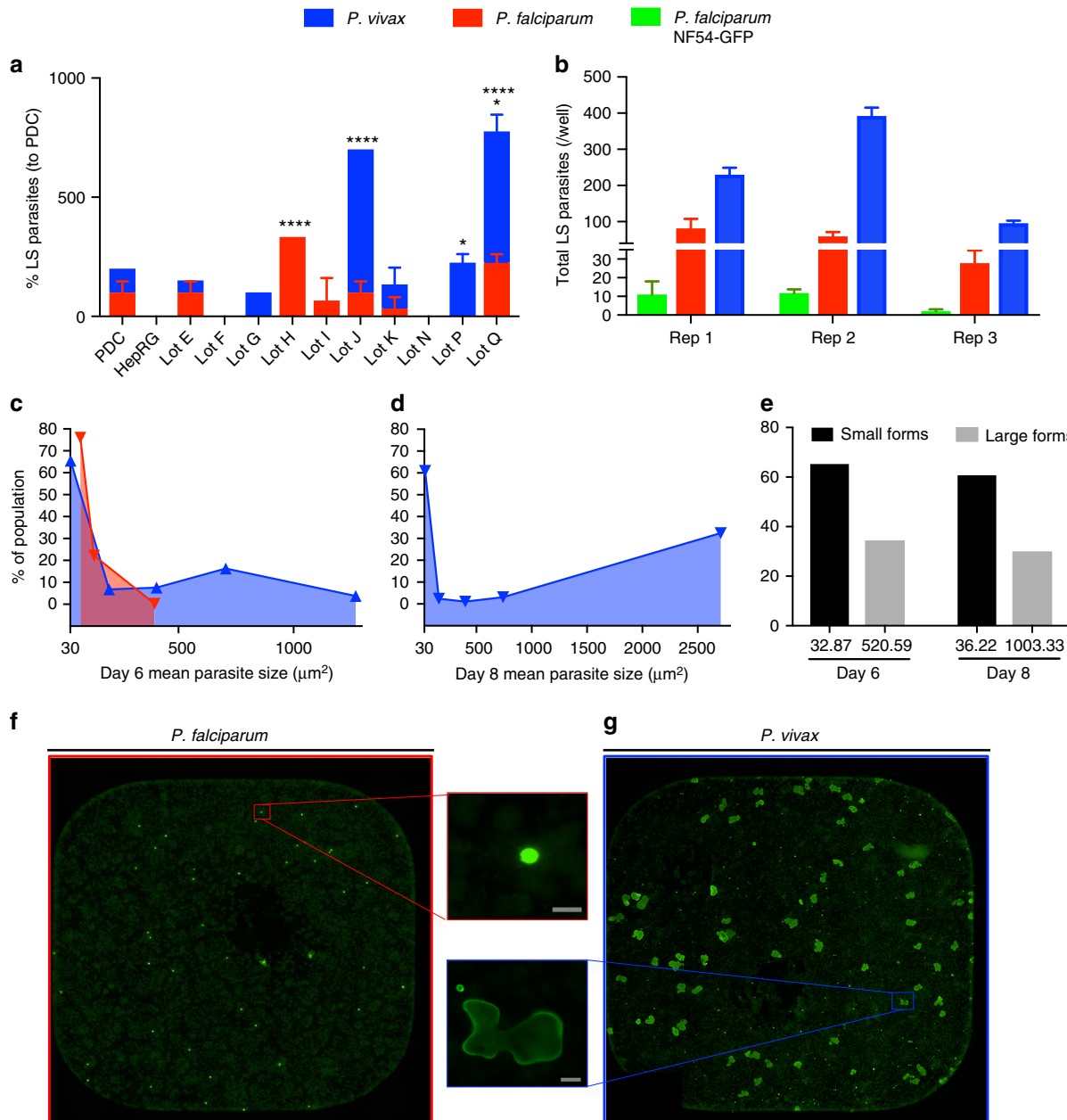

**Fig. 3** Characterizing *Plasmodium* developmental phenotypes in the primary human hepatocyte (PHH) culture system. **a** A total of ten cryopreserved PHH donor lots were screened for infectability and liver stage (LS) development by cryopreserved *P. vivax* and *P. falciparum* sporozoites in comparison to donors PDC and differentiated HepaRG line. Four donors (lot H, J, P, and Q) were significantly higher in percent of LS parasites relative to PDC while two donors (lot F and N) showed no LS parasite growth. Alternatively, specific donors appear to only be susceptible to a single *Plasmodium* species (lot G and P to *P. vivax*, lot H and I to *P. falciparum*). **b** *Plasmodium* sporozoites were added to PHHs at 5000 sporozoites per well with *P. vivax* field isolates showing highest LS development rates between 2 and 8.3%, *P. falciparum* NF54-WT between 0.6 and 2%, and *P. falciparum* NF54-GFP between 0.04 and 0.4%. **c** On day 6 of LS development, *P. falciparum* has a relatively synchronized growth with a large parasite population at a mean size of >100 μm². **d, e** *P. vivax* LS parasites have a larger distribution of size from day 6 to day 8 (**d**) with a consistent hypnozoite (~60%) to developing schizont (~40%) ratio in the total population (**e**). **f, g** Representation of a single 384-well view showing day 6 *P. falciparum* LS schizonts stained with anti-GAPDH (left) and day 8 *P. vivax* developing LS schizonts and dormant hypnozoites stained with anti-rUIS4 (right) imaged on the Operetta HCI system, Perkin Elmer. Graph bars represent means with s.d. from an independent experimental replicate ($n = 3$, **a**) or for biological replicates ($n = 3$) with experimental replicates ($n = 3$, **b**). Statistical significance was determined using a two-way ANOVA followed by Dunnett's multiple comparisons to PDC (**a, b**) where significance is represented by $P < 0.05$ (*) and $P < 0.0001$ (****). *Plasmodium* spp. mean parasite size distribution (**c–e**) was calculated from biological replicates ($n = 3$) with experimental replicates ($n = 26$, *P. vivax*) or ($n = 8$, *P. falciparum*). Gray scale bars represent 10 μm

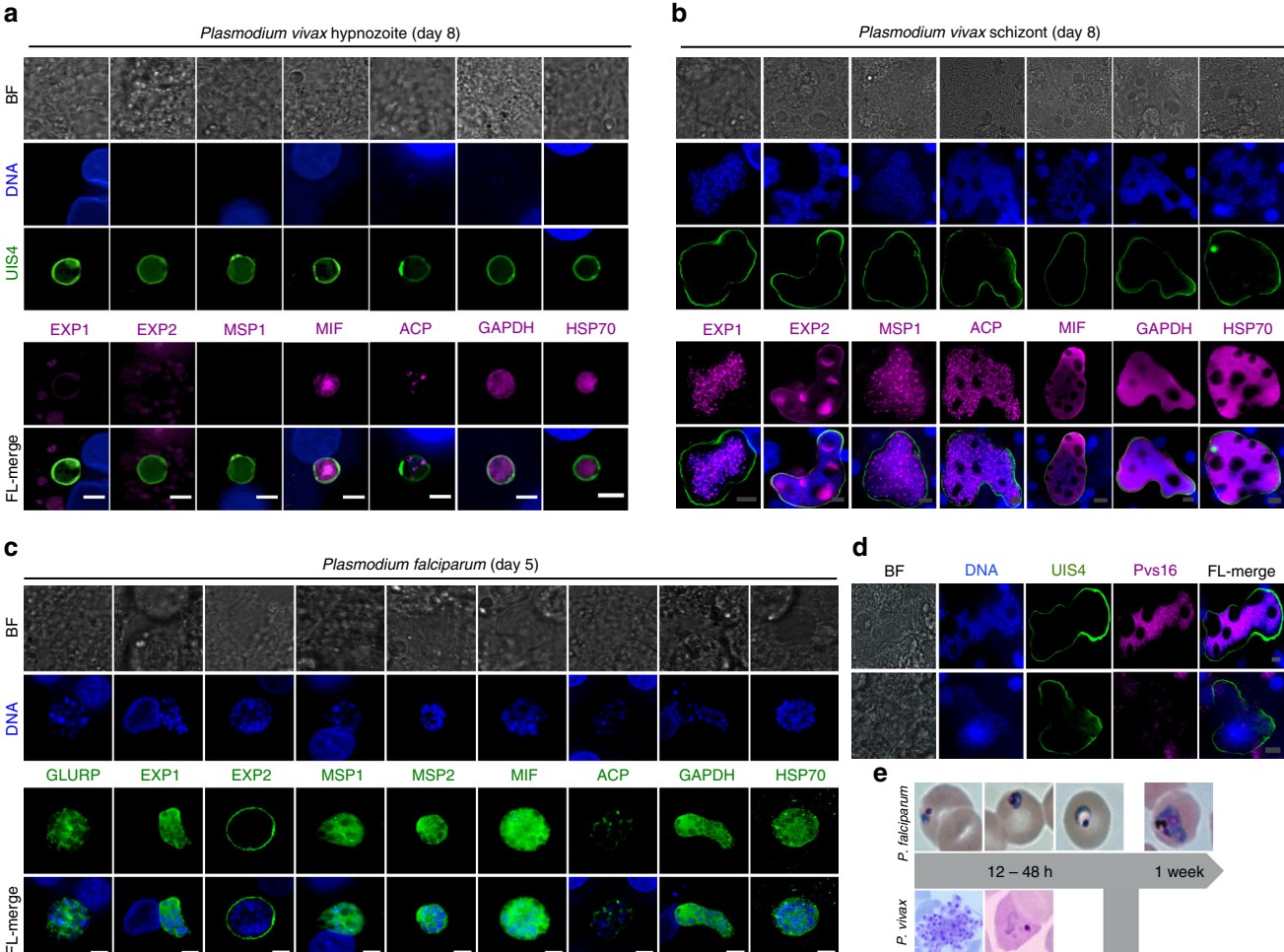

**Fig. 4** High-resolution immunofluorescent identification of *Plasmodium* liver stage (LS) parasites. **a** High-resolution images of *P. vivax* hypnozoites demonstrate these forms have minimal nuclear material and are negative for schizogony markers EXP1, EXP 2 and MSP1. Hypnozoites stain positive for cytosolic markers MIF, HSP70, and GAPDH and reveal a functioning apicoplast. **b** By day 8, *P. vivax* schizonts are several times larger than the host cell hepatic nucleus, feature genome replication and segmentation, and stain positive for EXP1, EXP2, ACP, and MSP1. **c** The LS of *P. falciparum* is shorter than that of *P. vivax* schizonts; developing parasites are correspondingly less small. By day 5 merozoite segmentation has begun (as noted by ACP staining of separate apicoplasts) but not complete (as noted by diffuse staining of MSPs). **d** Immunofluorescent staining of day 8 *P. vivax* LS schizonts with anti-Pvs16, a sexual stage-specific biomarker for immature gametocytes, showed co-localization with developing LS merozoites indicated by segmented DNA. However, anti-Pvs16 signal does not appear in every LS. **e** The PHH system successfully supports complete maturation of *Plasmodium* LS schizonts measured by breakthrough into blood stage using reticulocyte (*P. vivax*, days 9–11) or RBC (*P. falciparum*, days 7–8) overlays with initial giemsa staining every 6 h. *P. vivax* overlays show formation of merozoite packages which rupture into the reticulocyte culture leading to invasion. Alternatively, no merosomes were captured in *P. falciparum* overlays but early rings were present within the first 12 h and continued culture progressed to an asynchronous population at >1% parasitemia. White scale bars represent 5 μm, gray scale bars represent 10 μm

targeting proteins expressed at different stages of maturity (Fig. 4a–c). The *P. vivax* hypnozoites and schizonts were distinguishable by size beginning on day 4 post-infection with distinct individual merozoites evident in mature day 8 schizonts. In addition to the small sizes, hypnozoites stained positively for Upregulated In Infectious Sporozoite-4 (UIS4), a parasitophorous vacuole membrane (PVM) protein as previously noted, but were negative for both PVM markers Exported Protein-1 (EXP-1) and Exported Protein-2 (EXP-2) which function in active export in developing LS schizonts (Fig. 4a)[31,38,39]. Notably, day 8 *P. vivax* LS schizonts showed an accumulation of EXP-2 protein in unknown vacuoles suggesting a role in degradation or sequestration (Fig. 4b). All multinucleated developing stages of *P. vivax* and *P. falciparum* were stained positively for Acyl Carrier Protein (ACP) and Glyceraldehyde-3-Phosphate Dehydrogenase (GAPDH), providing evidence of active fatty acid biosynthesis and glycolysis[39]. In addition, mature LS schizonts had high

cytoplasmic expression of Macrophage Migration Inhibitory Factor (MIF) and Heat Shock Protein 70 (HSP70) confirming these parasites expressed proteins to modulate the human host (Fig. 4b, c)[31]. Mature hypnozoites stained negative for Merozoite Surface Protein 1 (MSP1) whereas day 8 LS schizonts showed MSP1 surface localization on segmenting merozoites (Fig. 4a). Day 5 *P. falciparum* LS schizonts were positive for both MSP1 and MSP2, even though the diffuse cytoplasmic localization indicated merozoite segmentation was in process (Fig. 4c)[40,41]. Interestingly, a portion of *P. vivax* late-stage schizonts stained positive for Pvs16, a gametocyte-specific protein, suggesting some LS merozoites are pre-committed as first generation gametocytes prior to the erythrocytic cycle (Fig. 4d and Supplementary Fig. 7). Finally, to confirm complete LS parasite maturation in the 384-well PHH culture system, merozoite infectivity was evaluated by addition or overlays of highly enriched human CD 71[+] reticulocytes to *P. vivax* LS cultures (days 9–11) and fresh human

**Table 1 High-content imaging and analysis of *Plasmodium* liver stage (LS) parasites**

| Parasite population | Size category | Area | Mean intensity[a] | Maximum intensity[a] | Cell roundness |
|---|---|---|---|---|---|
| Small forms: | | | | | |
| 1 | 1 | >15 μm$^2$<100 μm$^2$ | >1500 | >3000 | >0.80 |
| Large forms: | | | | | |
| A | 2 | >99 μm$^2$<300 μm$^2$ | >800 | >1500 | >0.70 |
| B | 3 | >299 μm$^2$<500 μm$^2$ | >250 | >1000 | >0.70 |
| C | 4 | >499 μm$^2$<1000 μm$^2$ | >250 | >1000 | >0.70 |
| D | 5 | >999 μm$^2$<2500 μm$^2$ | >200 | >300 | >0.60 |
| E | 6 | >2499 μm$^2$ | >200 | >600 | >0.38 |

Defined fluorescent imaging parameters and thresholds used to identify and quantify *Plasmodium* LS parasites using Perkin Elmer Harmony software. Small forms refer to *P. vivax* hypnozoites (mean ~30 μm$^2$) but with wide size range also represents for *P. falciparum* day 6 schizonts (mean ~80 μm$^2$). Large forms refer to developing schizonts and are further categorized into 5 distinct groups based upon area and intensity
[a] Based on FITC channel

red blood cells (RBCs) to *P. falciparum* cultures (days 7–8). Within first 24 h of *P. vivax* overlays, merozoite packages (merosomes) were seen released into the collected media along with *P. vivax*-infected reticulocytes and although we did not attempt to observe *P. falciparum* merosomes, early ring-infected RBCs were detected and continued development within RBCs was observed (Fig. 4e).

**Design consideration for treatment mode and HCI endpoint.** Ideally, a *P. vivax* and *P. falciparum* PE therapeutic screening platform should capture anti-parasite inhibitory activity against sporozoite migration, hepatocyte infection, LS development, and blood-stage breakthrough. To achieve these objectives, assay designs were considered either prophylactic, to prevent infections, or radical cure, to eliminate existing infections (Fig. 1b). For evaluating prophylactic efficacy of vaccines, sporozoites were exposed to immune antibodies or sera at the time of or just prior to their addition to the hepatocyte cultures. Similarly, prophylactic drugs were evaluated for activity by exposure at the time of sporozoite infection. However, LS assays for *P. vivax* require evaluation for radical cure activity to eliminate mature hypnozoites, as early drug application (prior to day 4–5 post-infection) can be effective on immature hypnozoites but inactive on mature hypnozoites (day 5 and beyond)[42]. Regardless of treatment mode, all cultures were fixed prior to release of mature merozoites to enable HCI-based quantification of growth using our experimentally defined LS phenotypes based on size and differential protein expression (UIS4 and GAPDH) (Table 1). In our 4-step HCI-based screening assays, antibody reagents were chosen for their staining patterns to facilitate automated quantification by defining the full LS form to enable measurement of growth area by automated morphometric analyses. Quantified populations were then normalized to controls and dose response curves fitted to generate half-maximal effective concentration (EC$_{50}$) and inhibitory concentration (IC$_{50}$) in Collaborative Drug Discovery (CDD) Vault or Graphpad Prism (Fig. 5). Automated imaging and quantification on an Operetta or ImageXpress was highly robust with 95–100% accuracy in identifying developing LS schizonts and hypnozoites when compared to conventional manual techniques (Table 2). Overall, automated HCI rapidly increased screening throughput allowing at least three 384-well plates to be analyzed within 24 h on an Operetta using wide field mode and 8 h on an ImageXpress spinning disc confocal imaging system.

**Anti-CSP antibodies show species-specific inhibition.** Next, our LS platform was validated for inhibition of liver stage developmental assays (ILSDA; Fig. 1b) by examining specificity and sensitivity of well-characterized monoclonal antibodies (mABs), anti-PfCSP mAB 2A10, anti-PvCSP mAB 2F2, and PvCSP mAB 2E10.E9[43–47]. Several ILSDA protocol optimizations lead to improved experimental results. First, we discovered that sporozoites of both species exposed to mABs in phosphate buffered saline (PBS) adversely affected ILSDA outcomes by significantly decreasing sporozoite viability, hepatocyte infection rates, and increasing false positives; therefore, a buffered cell culture medium (RPMI or hepatocyte culture media) was used for incubation with mAB (Supplementary Fig. 8a, b)[24,45,48–50]. Second, having found sporozoite invasion rates were improved by allowing the sporozoite–hepatocyte interaction to persist for 24 h instead of the conventional 3-h interaction followed by a wash step, ILSDAs were performed by co-incubation of sporozoites, mAB, hepatocytes, and media overnight instead of the typical sporozoite pre-treatment with mAB. Third, anti-PvCSP mAB 2F2 exhibited a minimal difference in IC$_{50}$ values for sporozoites exposed to the mAB for 20 min prior to addition to hepatocytes vs. co-incubation of mAB, sporozoites, and hepatocytes (Supplementary Fig. 8c). Finally, the assay endpoint was extended to day 6 or 8 in order to identify potential late-stage developmental phenotypes that are not easily evaluated with current LS platforms using hepatoma cells.

Our HCI analysis at 6 days post-infection confirmed that anti-PvCSP mAB 2F2 completely inhibited *P. vivax* sporozoite invasion and LS development at concentrations of 250 μg ml$^{-1}$ with an IC$_{50}$ of 4.60 μg ml$^{-1}$ ($n = 3$, $n' = 2$) (Fig. 6a). This inhibition was specific, as *P. falciparum* sporozoites exposed to anti-PvCSP mAB 2F2 resulted in normal *P. falciparum* LS development and *P. vivax* sporozoites exposed to anti-PfCSP mAB 2A10 resulted in normal *P. vivax* LS development. These results were expected as anti-*P. falciparum* and *P. vivax* sporozoite immunity (natural immunity or by immunization with irradiated sporozoites) is directed towards the dominant Bc epitopes of the central repeat regions of mammalian CSPs, which are the epitope targets of these anti-CSP mAbs. Similar analysis of *P. vivax* sporozoite attenuation with anti-PvCSP mAB 2F2 revealed a novel discovery that *P. vivax* LS parasites were significantly reduced in size and had lower DNA content, indicating post hepatocyte-invasion antibody inhibition of LS development (Fig. 6b, c). Moreover, additional day 8 ILSDAs performed with cryopreserved *P. vivax* VK247-positive sporozoites and anti-2E10.E9 mAB (targeting CSP subtype VKS247) obtained a similar IC$_{50}$ (2.85 μg ml$^{-1}$) compared to ILSDAs with fresh *P. vivax* sporozoites, indicating a possible use of cryopreserved sporozoites for standardized screening (Supplementary Fig. 9)[51,52]. While further study is needed, these results show our platform is suitable for gaining a deeper

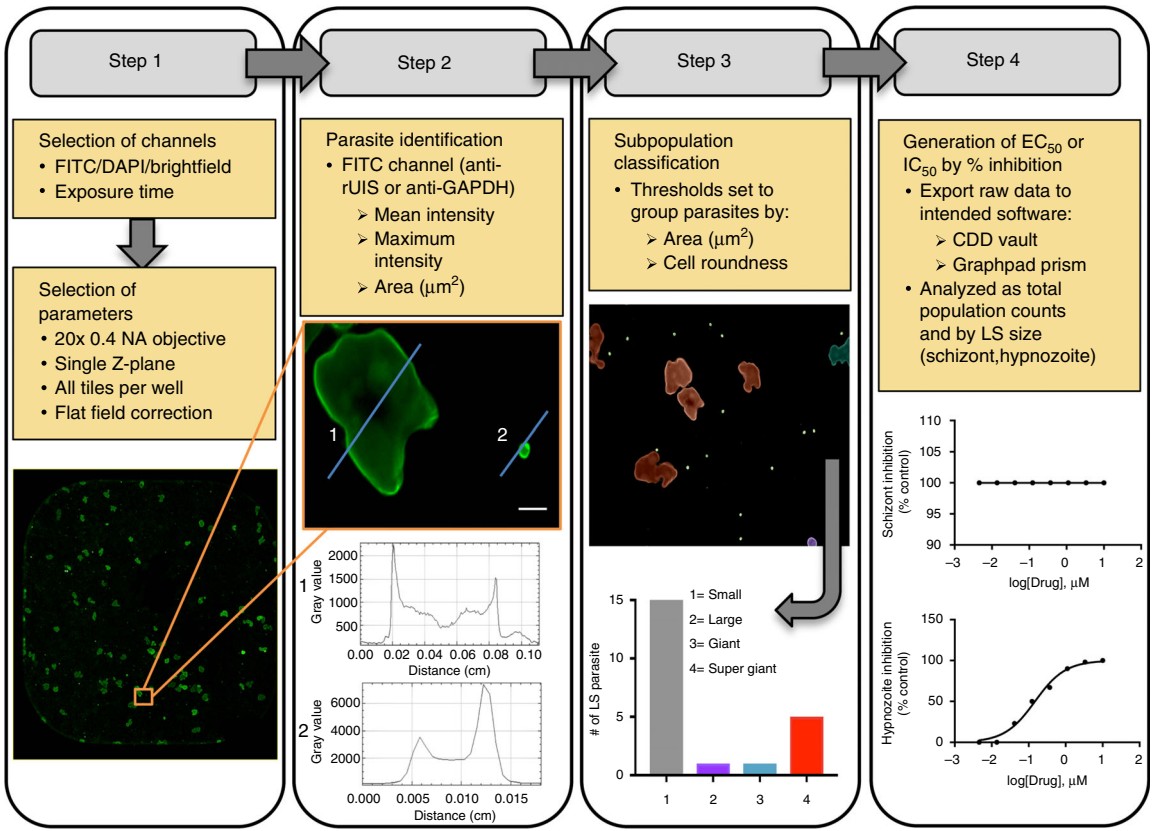

**Fig. 5** Enhanced-throughput imaging and analysis for PE therapeutic screening. Schematic represents automated imaging and data analysis work flow using the Operetta high-content imaging (HCI) system (Perkin Elmer) where four consecutive steps are followed for maximum throughput of liver stage (LS) screening. Step 1: fluorescent channels are selected based on secondary conjugates and imaging parameters are set based on ×20 magnification (0.4 NA). Step 2: LS parasites are identified by high intensity staining patterns of LS-specific antibody (i.e., anti-rUIS4 for *P. vivax* or anti-GAPDH for *P. falciparum*) denoted by line plot profile (1, 2) Step 3: LS parasites are further characterized into size categories by area and roundness to ensure software is correctly selecting the entire parasite. Step 4: After LS parasites are quantified, raw data is imported into CDD vault or Graphpad Prism where % inhibition and dose-response curves are calculated based on total parasites and size. *P. vivax* LS assays specifically determine compound activity on hypnozoite population. White scale bars represent 5 μm

**Table 2 Comparison of *P. vivax* liver stage (LS) parasite quantification: Operetta vs. Manual**

| | Harmony software | | Manual count | | Accuracy | |
|---|---|---|---|---|---|---|
| | Small forms | Large forms | Small forms | Large forms | Small forms | Large forms |
| well 1 | 180 | 266 | 179 | 252 | 99.40% | 99.40% |
| well 2 | 182 | 249 | 174 | 243 | 95.40% | 97.50% |
| well 3 | 180 | 256 | 176 | 256 | 97.70% | 100% |

Perkin Elmer Harmony software exhibits high level precision of identifying *P. vivax* LS parasites with >95% accuracy

understanding of the activity of anti-CSP antibodies as we identified and quantified inhibitory effects manifesting during and after hepatocyte invasion.

Similar species-specific concentration-dependent results were obtained for *P. falciparum* sporozoite infection and development in PHHs using anti-PfCSP mAB 2A10. Conversely, anti-PvCSP mAB 2F2 had no measurable effect on *P. falciparum* development at any concentration (Fig. 6d). The higher concentrations of anti-PfCSP mAB 2A10 of 80 and 40 μg ml$^{-1}$ led to a 100% functional inhibition of parasite development with elevated reduction (>50% inhibition) in parasite development across all remaining concentrations, respectively (Fig. 6d). Similar to the *P. vivax* ILSDA,

*P. falciparum* sporozoites exposed to lower concentrations of antibody (10 μg ml$^{-1}$) permitted hepatocyte infection but resulted in late-stage LS developmental growth defects and delayed maturation (Fig. 6e, f). This consistently observed growth attenuation activity suggests that in addition to invasion inhibition, an anti-sporozoite antibody can attenuate intracellular LS development through an unknown mechanism.

To determine the utility of the ILSDA to experimentally evaluate functional immunogenicity of novel vaccines, *P. vivax* and *P. falciparum* sporozoites were treated with mouse immune sera samples (G2, G3, G6, and G7) raised against a self-assembling protein nanoparticle (SAPN) (FMP014/ALF, FMP014/ALFQ, FMP014V/ALF, and FMP014V/ALFQ, respectively)[53]. Samples G2 and G3 were derived from immunization with the FMP014 monomer containing PfCSP CD4$^+$ and CD8$^+$ epitopes, universal $T_H$ epitopes, portions of the α-TSR domain, and 6 repeats of the NANP motifs of the PfCSP with differing Army Liposomal Formulation (ALF) based adjuvants (ALF or ALQ)[53]. Samples G6 and G7 were derived by immunization with a SAPN assembled from an FMP014V monomer containing two copies of the PvCSP (VKS type 210) repeat region motif and the α-TSR domain of PfCSP[53]. Invasion and development of *P. falciparum* sporozoites were inhibited in a dose-dependent manner similar to the anti-CSP mAB (Fig. 6g) for all four serum samples with G2 exhibiting the strongest inhibitions for all dilutions and a growth phenotype at 1:16 dilution. Alternatively,

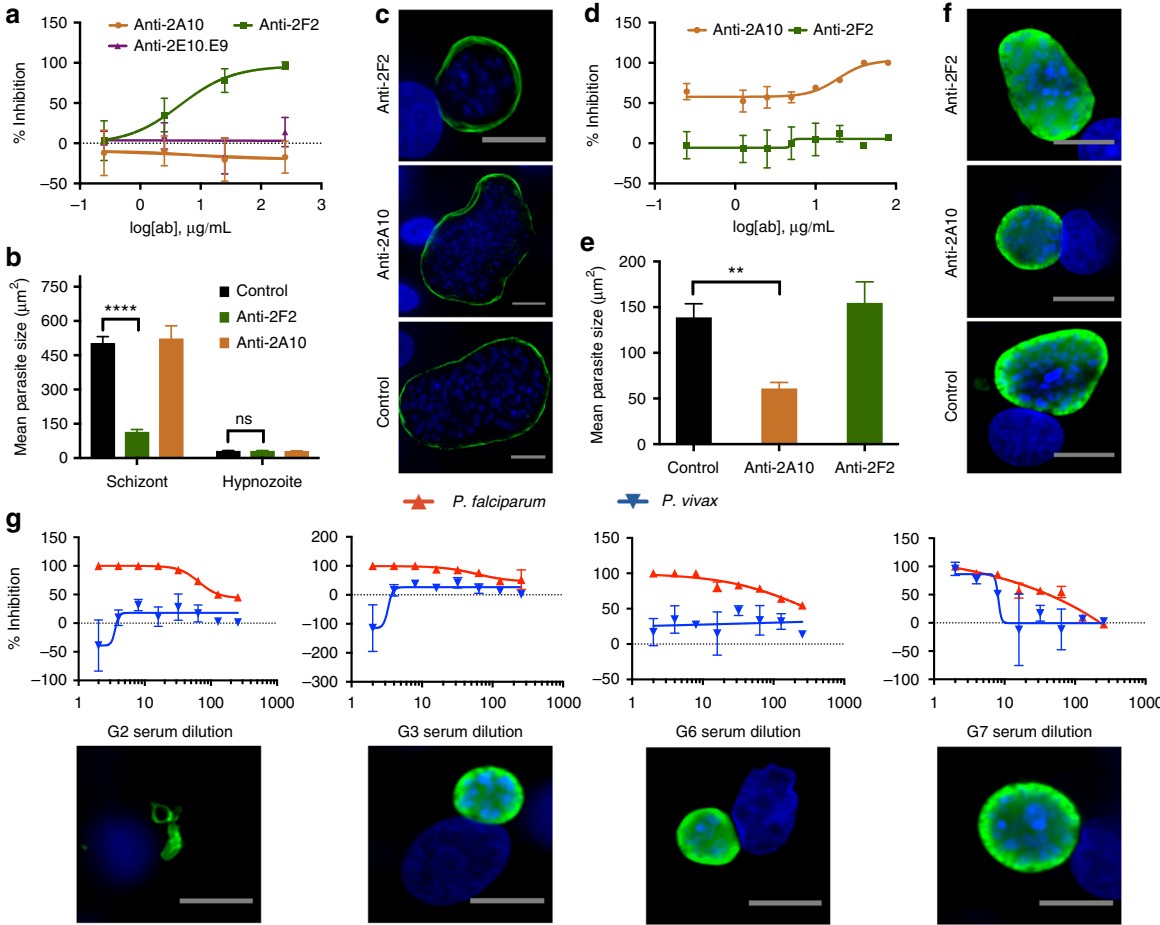

**Fig. 6** *Plasmodium falciparum* and *P. vivax* inhibition of liver stage development assays (ILSDAs). **a** *P. vivax* sporozoites exposed to anti-*P. vivax* circumsporozoite protein monoclonal antibody 2F2 (PvCSP mAB 2F2) showed a concentration dependent dose response with complete inhibition (100%) of sporozoite invasion and LS development at 250 µg ml⁻¹. **b**, **c** Day 6 developing *P. vivax* liver stage (LS) parasites showed a decreased growth (µm²) phenotype to the 25 µg ml⁻¹ PvCSP mAB 2F2 concentration compared to the no antibody control. **d** *P. falciparum* sporozoites exposed to anti-*P. falciparum* circumsporozoite protein monoclonal antibody 2A10 (PfCSP mAB 2A10) showed complete inhibition (100%) at concentrations 80 and 40 µg ml⁻¹ with >50% inhibition at remaining concentrations. **e**, **f** Similarly, day 6 *P. falciparum* LS parasites had a decreased growth phenotype at the 10 µg ml⁻¹ PfCSP mAB 2A10 concentration compared to the no antibody control. **g** Dose response ILSDAs for test sera immunized against nanoparticles G2, G3, G6, and G7 (top), and representative images of day 6 *P. falciparum* LS parasite forms exposed to a 1:16 serum dilution (bottom). Graph bars represent means with s.d. biological replicates ($n = 3$, **a**, **b** and $n = 2$, **d**, **e**, **g**) with experimental replicates ($n = 2$, **a**, **b**, **d**, **e**, **g**). Statistical significance was determined for using two-way ANOVA followed by Tukey's multiple comparisons to control where values are represented as $P < 0.0001$ (****) and no significance (ns, **b**) or was determined using one-way ANOVA followed by Dunnett's multiple comparisons to control where values are represented by $P < 0.01$ (**) and no significance (ns, **e**). Gray scale bars represent 10 µm

sample G6 only moderately inhibited *P. vivax* sporozoites and G7 showed 100% inhibition at lowest dilutions with an IC₅₀ at 1:8 dilution ($n = 2$, $n' = 2$) (Fig. 6g). The pre-bleed serum control showed minimal inhibitory effect compared to the unexposed sporozoite control (Supplementary Fig. 8d). To overcome experimental bias, all percent inhibition calculations were standardized to the pre-bleed serum control[48].

**Compound library screen targeting LS parasites**. Several compound collections were screened for activity against *P. vivax* LS parasites in our LS platform following the drug treatment modes described in Fig. 1. Compounds from the MMV portfolio were tested in eight-point dose response assays, including triplicate controls (KDU691 and atovaquone) in addition to 36 compounds per plate, with concentrations ranging from 10 µM to 5 nM represented in singleton wells. In every assay plate, potent activity of phosphatidylinositol 4-kinase inhibitor (PI4K) KDU691 was confirmed against all forms except hypnozoites treated in the

radical cure mode and atovaquone activity only against schizonts treated in prophylactic mode (Fig. 7a). To demonstrate multi-species capability and cross-species comparisons, a small subset of 6 MMV compounds were evaluated in the eight-point dose response assay against *P. falciparum* LS schizonts (Fig.7b and Supplementary Table 5). Compounds were tested in duplicate in two independent experiments in prophylactic mode only, as *P. falciparum* does not produce hypnozoites and completes development prior to the radical cure treatment times. Comparison of EC₅₀ values between *P. vivax* and *P. falciparum* revealed similar efficacies between atovaquone, DSM421, P218, and KAF156. Pyrimethamine showed a potent EC₅₀ at 0.026 µM for *P. falciparum* and was confirmed inactive against *P. vivax* when assayed at a single concentration of 10 µM[54,55].

Next, a set of 913 repurposed compounds from the Calibr Bioactive Library were tested at 10 µM single concentration in the radical cure treatment mode. Compounds with schizonticidal or hypnozonticidal activity were identified, including one compound, monensin, showing complete activity against both forms

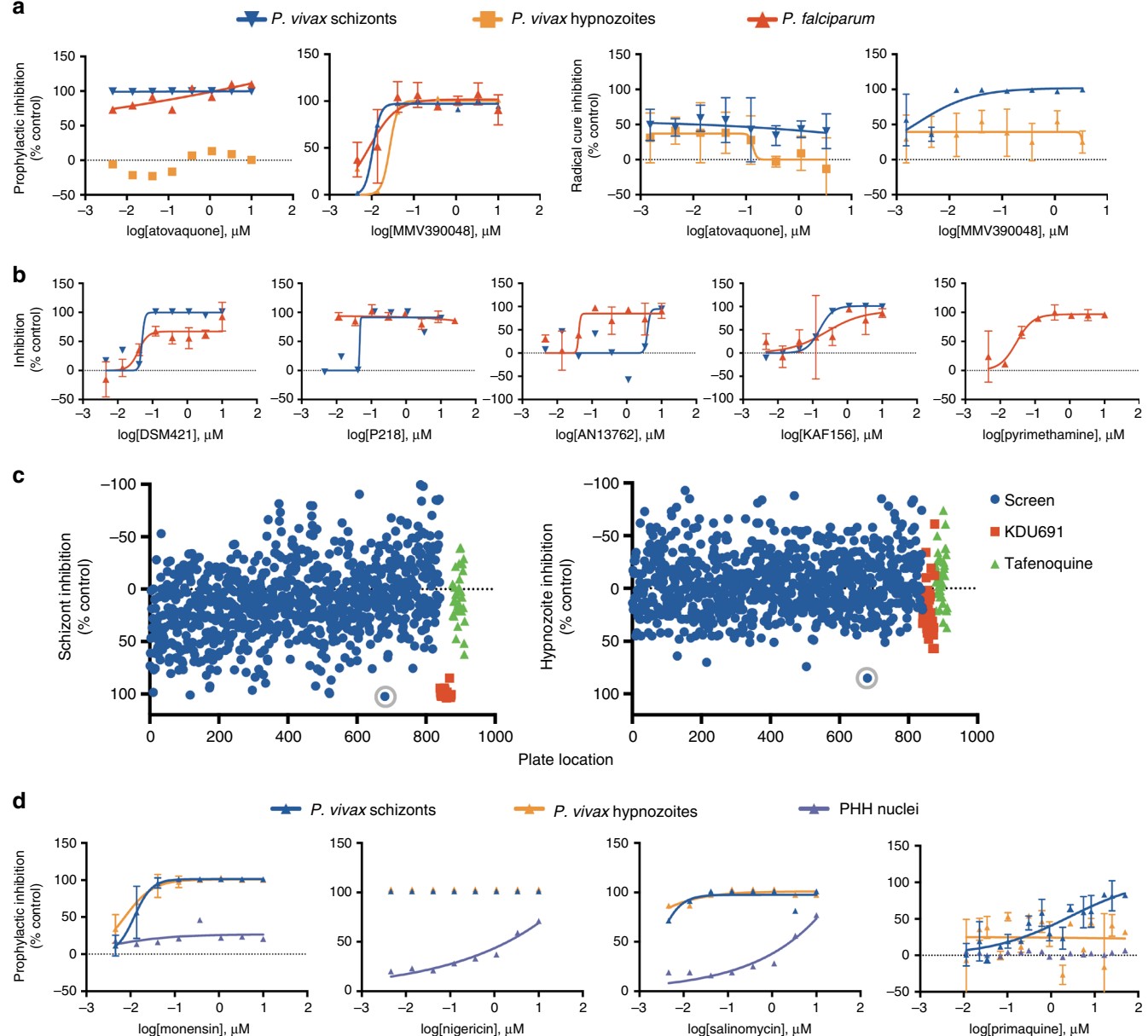

**Fig. 7** Medium throughput dose response and single point antimalarial compound screening. **a** Dose response charts of inhibition of *P. vivax* and *P. falciparum* LS development following treatment with control compound KDU691 (a PI4K inhibitor) and reference compound atovaquone. **b** Dose response curves for six MMV portfolio drugs against both *P. vivax* and *P. falciparum* LS schizonts. **c** Activity of 913 Calibr Bioactive compounds against *P. vivax* in radical cure mode; activity for each compound is indexed on the X axis by plate position with individual plate control wells grouped and indicated. Monensin, active against *P. vivax* hypnozoites and schizonts in both prophylactic and radical cure modes, is indicated within a gray circle. **d** Dose response plots of inhibition of *P. vivax* LS parasites following treatment with three different ionophores or primaquine; PHH nuclei counts, normalized to DMSO controls, are shown to demonstrate selectivity index. Graph bars represent means with s.d. of biological replicates (*n* = 2) for DSM421, P218, AN13762, KAF156, pyrimethamine, and monensin; biological replicates (*n* = 3) for MMV390048 and atovaquone; or biological replicates (*n* = 4) for primaquine

(Fig. 7c, Supplementary Table 6). Monensin activity was more potent and rapidly clearing compared to 8-aminoquinolines, primaquine, and tafenoquine, which were negative for activity after 3 days of treatment in all assay modes except primaquine against schizonts following prophylactic treatment (Fig. 7d). A radical cure dose-response assay was performed to confirm the ability of monensin to clear hypnozoites and was found to be effective with an EC$_{50}$ of 398 nM (*n* = 2, *n′* = 3). Using our prophylactic 8-point dose response assay, we explored the efficacy of three other ionophores: nigericin, salinomycin and lasalocid-A[56],[57]. All ionophores were found to potently inhibit all *P. vivax* LS development. Because many ionophores are toxic to specific

species, we compared PHH nuclei counts to better understand ionophore selectivity index (Fig. 7d and Supplementary Table 6)[58]. Nigericin, in particular, cleared the culture of all parasite forms at all doses tested (EC$_{50}$ < 5 nM) while hepatic toxicity was only noted at much higher doses (>100 nM), demonstrating a selectivity index of over 20. A subset of compounds were assayed in three different PHH lots, NLX, PDC, and UBV to identify host-cell-specific effects on efficacy. Active compounds atovaquone and MMV390048, a PI4K inhibitor, were found similarly active across all donors, but prodrug primaquine was found active only in lots with higher CYP2D6 activity (Supplementary Table 2, Supplementary Fig. 10).

## Discussion

The lack of efficient high-throughput screening assays has been a major constraint limiting development of much needed new therapies targeting *P. vivax* and *P. falciparum* PE stages. The in vitro liver model reported here removes this barrier by significantly reducing the use of precious biological materials; each individual assay unit requires only $1.0 \times 10^4$ hepatocytes and only $5.0 \times 10^3$ *P. vivax* or *P. falciparum* sporozoites, representing a >20-fold reduction per assay unit relative to current platforms[28–30,50]. When extrapolated for a full 384-well plate run at least $2.0 \times 10^6$ sporozoites are required per plate, a practical number to obtain from dissection of salivary glands from 20 to 200 mosquitoes. Nonetheless, this small cell culture surface area of only 10 mm$^2$ typically supported >200 *P. vivax* or >50 *P. falciparum* LS parasites per well, exceeding the averages of infected hepatocytes obtained in traditional models using larger 96-well plates[59,60].

We attribute our improved sporozoite isolation and infection techniques, as well as access to properly cryopreserved hepatocytes as the key factors responsible for the greatly improved infection rates. Following characterization of 13 different PHH lots in addition to lot PDC and HepaRG, several lots from were found highly viable, monolayer-forming, and functional; all characteristics ideal for *Plasmodium* LS development (Supplementary Tables 2 and 3, Supplementary Fig. 5). However, the specificity of PHH lots for either *P. falciparum* or *P. vivax* may be better attributed to the hepatic phenotypes. Of all the PHH donor lots examined for sporozoite invasion and LS development, some lots were not supportive of either or both *P. vivax* and *P. falciparum*. We hypothesize this bimodal support of infection in otherwise healthy hepatocytes is a product of natural variation in presentation of hepatic surface receptors CD81 and SR-BI, as *P. vivax* and *P. falciparum* hepatocyte entry has been found receptor-specific[61]. While relative expression of surface receptors could not be analyzed in this study, additional proteomic and transcriptomic tools may be needed to identify biomarkers linked to a compatible donor phenotype. Alternatively, this variation could result from the cryopreservation process possibly leading to hepatocyte membrane receptor loss or undermining hepatic fitness to support intracellular parasites. Our hepatocyte bio-image assessment of the different PHH donor lots did suggest a strong link between enhanced sporozoite invasion and LS development to highly dynamic mitochondrial activity and increased bile canaliculi network formation. This indicates that the quality of hepatocyte lot may be more important than variation among individual donors. The mitochondria in hepatocytes are essential for survival, with main roles in energy metabolism (ATP), ion homeostasis, and apoptosis. Hepatic mitochondrial activity is likely vital in LS development, but host-parasite interactions are still largely uncharacterized and further investigation is needed[62,63]. Regardless, we demonstrate the advantage of using a pre-validated, effectively cryopreserved PHH lots over cell lines as PHHs support higher invasion and development rates and full LS development during long-term radical cure vaccine and drug assays. Until more rapid phenotype assessment tools are available, investigators using PHHs for LS assays will first need to screen multiple lots to find the ones most suitable for their species of interest. Fortunately, with up to 1000 aliquots of cryopreserved PHHs resulting from a production run, many thousands of individual assay units can be experimentally analyzed with the same genotypic and phenotypic background of a single suitable hepatocyte lot after identification.

Equally important to reducing the rate-limiting requirements were mitigating costs and availability of materials and automation. We estimated the cost of consumable reagents (plate, hepatocytes, medium) were less than US ~$2 per well and materials could be readily shipped to assay locations. The live portion of the assay required no robotic automation, enabling rapid implementation of the protocol at multiple laboratories including those in malaria-endemic countries. This flexibility was and will continue to be essential for developing new antimalarial PE therapies, especially against vivax malaria, since blood stages of *P. vivax* cannot be continuously cultured making widespread production of gametocytes for PE assays in nonendemic countries a major obstacle[59]. While shipment of *Plasmodium* spp.-infected mosquitoes from endemic countries to screening facilities is possible and was performed to generate some of these data described here, we note far greater throughput can be achieved by moving the assay itself to the endemic country to be supported by sporozoites produced from local infections. As malaria control increases worldwide, continued efforts to screen for PE therapeutics could be enabled by cryopreserved sporozoites, a concept also demonstrated in this study (Fig. 3a, Supplementary Fig. 4a, b. and Supplementary Fig. 9). Although direct comparisons were not made, our results surprisingly suggest laboratory strains of *P. falciparum* have reduced fitness in an in vitro LS assay.

The staining patterns with the panel of immuno-reagents used to characterize the LS morphologies in our in vitro cultures were similar to that noted in in vivo humanized mouse models, with either *P. vivax* or *P. falciparum*, confirming complete LS development in vitro[31,32]. In both mice and our in vitro culture, LS schizonts exhibit several large circular vacuoles of regular size, each about 1/20–1/10 the parasite total 2D area as observed under microscopy. The role and contents of these vacuoles are unknown, but high-resolution imaging revealed the membrane limiting the vacuole contained EXP2 and not UIS-4 and the internal void lacked MIF. Based on these observations, these uncharacterized vacuoles are likely a formation of the parasite membrane and not the parasitophorous vacuole membrane and would be separate from the parasite's cytoplasm. As *Plasmodium* biology and morphology contains several well-characterized sequestration mechanisms (i.e., micronemes and dense granules for invasion protein machinery of sporozoites and merozoites, or the digestive vacuole for sequestration of toxic hemozoin produced during the erythrocytic stage), further characterization of the contents in these novel LS compartments could reveal interesting new drug targets such as resident solute pumps or possible toxic byproducts produced during maturation[60]. Morphological characterization also helps explain one of the unknowns of *P. vivax* lifecycle--the parasite is able to form infectious gametocytes in only days compared to over 2 weeks for *P. falciparum* and epidemiology studies show gametocytes are circulating within days of LS development[64,65]. We find a large proportion of completely mature *P. vivax* LS schizonts express gametocyte commitment marker Pvs16 such that all merozoites in the form are positive; while the remaining schizonts are negative for Pvs16. As commitment to gametocytogenesis has been described to begin in the asexual lifecycle producing the merozoites that will form a gamete, this stain pattern suggests whole sub-populations of *P. vivax* LS schizonts are producing gamete-forming merozoites instead of asexually replicating merozoites[66]. This mechanism, if confirmed, would help explain vivax transmission between an asymptomatic individual and the mosquito vector.

Key to the development of the next generation antimalarial toolbox, our platform is the first to perform extended ILSDA's (days 6 or 8) in PHHs revealing not only inhibitory effects but also growth attenuation of maturing *P. falciparum* and *P. vivax* LS parasites[67]. Our ILSDA offers solutions to inherent issues of traditional ILSDA methods through assay standardization and high infection rates, thus greatly increasing reproducibility and sensitivity[45,50,68]. We show immune serum raised against SAPN-produced antibodies to FMP014/ALF (G2) and FMP014V/ALF (G6) both inhibit invasion of *P. falciparum* sporozoites. This

finding is of high importance as FMP014V/ALF does not have PfCSP NANP repeats, indicating all inhibition comes from antibodies to the α-TSR region, thus potentially leading to longer-lived levels of protection[53]. Lastly, our model has the ability to screen late-stage-arresting genetically-attenuated sporozoite vaccines along with vaccine candidates targeting first generation LS merozoites entering into blood-stage.

Medium throughput small molecule screening was performed using our HCI endpoint to quantify the number and growth of LS schizonts and differentiate *P. vivax* schizonts from hypnozoites. Control compounds atovaquone and KDU691 behaved as expected following characterization of these compounds in vitro and in vivo[42,69]. We also confirmed cross-species activity of several preclinical candidate drugs to characterize the predictive value of our screen. To better understand the possible effect of host-specific metabolism, an assessment of compound efficacy in different PHH lots found no drastic PHH-specific differences for optimized compounds, including MMV390048 and atovaquone, but did find a lot-specific outcome with primaquine in relation to each lot's CYP2D6 activity and these lots' ability to produce the active primaquine metabolite (which cannot yet be directly quantified). In addition to differential host cell metabolism, measurement of primaquine efficacy is confounded by how quickly treated LS parasites clear from culture. Primaquine treatment in vivo has been shown to result in malformation, but not necessarily clearance, of LS parasites, while our radical cure assay was taken out to only 3 days post compound addition so as not to miss observation of schizonticidal compound activity[70]. To alleviate concerns of a false negative due to metabolism, dosing concentration and repetitions (maximum 10 μM administered 3 times) were optimized to provide excessive opportunity for metabolically labile compounds to act on LS parasites. Furthermore, the MMV compounds and Calibr Bioactive Library screened were all metabolism-optimized compounds, thus reducing the likelihood of a false negative due to metabolism. Despite these protocol adaptations to circumvent hepatic metabolism, we cannot rule out that our screen may miss quickly metabolized compounds. Application of our model to single point screening revealed several interesting hits, including the ionophore monensin, and suggests *P. vivax* schizonts and hypnozoites are sensitive to ion homeostasis. The ionophores will be best used as positive controls for radical cure assays rather than new drugs given the known problems with ionophore toxicity and poor selectivity index[71]. In the least, these hits demonstrate dead LS parasites can be cleared from PHH culture, and therefore drug activity can be detected, following a relatively short 3-day treatment. These ionophore hits also suggest ion homeostasis could be a viable target for anti-malarial drug discovery[72].

In conclusion, for decades, preclinical malaria research on PE stages has been based on cell lines and animal models, or secondary effects from other *Plasmodium* lifecycle stages. We validate and promote the use of a primary hepatocyte, 384-well plate-based screening of PE therapeutic targets as an excellent resource to provide biological data and conclusions that may not be garnered from hepatoma or murine models.

## Methods

**Seeding 384-well plates with human hepatocytes**. Individual wells of a 384-well plate (Cat No. 781091, Greiner, Monroe, NC, USA) were coated with rat tail collagen I (Cat No. 354236, Corning, New York, NY, USA) diluted to a final concentration of 5 μg collagen per cm$^2$ in 0.02 M acetic acid, incubated at 37 °C overnight to ensure adsorption and washed thrice with 1 × PBS. Alternatively, after experimental confirmation, pre-collagen coated plates (Cat No. 781956, Greiner, Monroe, NC, USA) were used for all compound screening and a selection of ILSDA studies. For ILSDA and drug studies, cryopreserved primary human hepatocytes (Cat No. M00995-P and F00995-P, donor lots PDC, UBV, NLX) and hepatocyte culture medium (HCM) (InVitroGro™ CP Medium) were obtained from

Bioreclamation IVT, Inc, (Baltimore, MD, USA). For donor screens, cryopreserved primary human hepatocytes were obtained from four companies (Cat No. M00995-P and F00995-P, donor lots E-Q, Bioreclamation IVT, Inc, Baltimore, MD, U.S.A), (Cat No. HUCSD, donor lots HUM4096A, HUM4074D, Triangle Research Labs LLC, Durham, NC, USA), (Cat No. 20-0047, donor HHF17006 lot RTA0006 and donor HHM01013 lot DFA0003, Yecuris™, Tualatin, OR, USA), (donor lot HW40, HW39, no label, Mass General Hospital, Boston, MA, USA) and maintained following manufacturer's protocols. For HepaRG studies, NoSpin HepaRG™, a commercially available, terminally-differentiated hepatic cell line, were seeded at 20,000 liver hepatocytes per well following the manufacturer's instructions (Cat No. NSHPRG, Triangle Research Labs LLC, Durham, NC, USA).

All media was supplemented with a final concentration of 50 μg ml$^{-1}$ penicillin-streptomycin, 100 μg ml$^{-1}$ of neomycin, and 10 μg ml$^{-1}$ of gentamicin. Cultures were kept on HCM (media containing serum) throughout studies as serum-containing media was found important for long-term viability. Cryopreserved PHHs were thawed following manufacturer's protocols and viability assessed by trypan blue exclusion (typically 70–99%), resulting in recovery of 5–9 million viable cells per vial. Thawed PHHs were then diluted in HCM to a concentration of 800–900 cells per μl with 20 μl cell suspension added to individual wells. The relative number of hepatocytes seeded per well was set to achieve a confluent monolayer. At 2 days post seed each well received a complete media change (40–50 μl), then media was changed per the experimental protocol (Fig. 1b).

**Hepatocyte functional assays**. Albumin and Factor IX production were measured by collecting media from hepatocytes (PDC) seeded wells at set time points and then storing samples at −80 °C until collection was completed. For albumin measurements, 12 wells (at 50 μl volume) were collected every 2 days for 30 days with media addition after time point removal. Albumin measurements (ng ml$^{-1}$) were collected using the Albumin (human) AlphaLISA Detection kit specific for the EnSpire Alpha screen (Cat No. AL294C, Perkin Elmer, Waltham, MA, USA) and the mean with s.d. ($n = 6$) were reported. Samples ($n = 3$) were quantified for Factor IX ELISA per manufacturer's protocols (Cat No. EF1009-1, AssayPro, St. Charles, MO, USA). Cyotchrome P450 3a4 induction was measured using CYP3A4 P450 Glo™ luciferin-IPA kit (Cat No. V9002, Promega, Madison, WI, USA). Hepatocytes were induced for 72 h with 25 μM Rifampicin (Cat No. BP26795, Fisher, Hampton, NH, USA) in plate media, or an equivalent volume of DMSO in uninduced controls, and luciferin signal measured by a luminance plate reader following the lytic version of the manufacturer's protocol. Basal enzymatic characterization was performed for all 14 BIVT donor lots and provided by Bioreclamation IVT, Inc (Baltimore, MD, USA). Metabolic assays were run in triplicate and results represent the average of the 3 samples (Supplementary Table 2 and Supplementary Fig. 5). Activity results were analyzed by HPLC-UV or LC/MS/MS validated procedures. Metabolite formation for all enzymes was measured after a 60 min. Cell incubation at 37 °C, 5% $CO_2$ resulting in a final concentration of 1 million cells per ml. The standard 60 min incubation was used for this assay.

**Live imaging and analysis of hepatobiliary formation**. A total of 6 time points (day 2, 4, 8, 14, 21, 30) with experimental replicate wells ($n = 3$, $n' = 3$) were selected to phenotypically assess the PHHs (donor lot PDC) seeded in the commercial 384-well plate (Fig. 2a). For the Bioreclamation IVT PHH donor screen, a total of 4 time points (day 2, 4, 8, 14) with experimental replicate wells ($n = 1$, $n' = 3$) were selected to phenotypically assess the PHHs seeded in the commercial 384-well plate (Supplementary Table 3). Wells were stained for 45 min at 37 °C and 5% $CO_2$ with staining solution diluted in phenol-free, serum-free RPMI containing 10 μM Hoechst 33342 (Cat No. H3570, Life Technologies, Carlsbad, CA, USA), 750 nM Tetramethylrhodamine (TMRM) (Cat No. T668, Life Technologies, Carlsbad, CA, U.S.A.) and 4 μM CellTracker™ Green CMFDA dye (Cat No. C7025, Life Technologies, Carlsbad, CA, USA). An Operetta high-content imaging machine (Perkin Elmer, Waltham, MA, USA) was used in wide field mode (×20 objective, 0.4 NA) to capture 35 tiles within each 384-well while simultaneously acquiring 4 channels (Transmitted, TRITC, FITC, DAPI) representing brightfield, respiration, transport, and viability. A z-plane stack of −2 to 10 μm with a 2 μm plane distance was acquired for each well. Using the Harmony software (version 4.1), a maximum image projection merging all z-planes was formatted for each well per time point. First, total hepatocytes per well were calculated by using Hoechst staining of hepatocyte nuclei where populations termed "healthy" and "unhealthy" were determined by area (μm$^2$) and MFI ≥ 3000. Next, the area and fluorescent intensity of TMRM was calculated using the emitted fluorescence (TRITC channel) where the MFI of entire well was used to compare mitochondrial activity between the time points.

To accurately find and classify the bile canaliculi as active or inactive transport, a combination of several building blocks and image filter sets were required using Harmony software (4.1) (Supplementary Fig. 2). Post compilation of the z-stack, a Gaussian image filter was applied to enrich contrast of active bile canaliculi and normalize background intensity. Simultaneously, the image was filtered selecting for intensity of the FITC channel allowing for identification of hepatocyte cells incapable of transporting dye. An image calculation was applied (Gaussian filtered image plus filtered FITC channel) and the following criteria were obtained for all classified bile canaliculi at all time points; morphology (length, width, area, roundness), and intensity (sum, mean, maximum, minimum). Thresholds to

distinguish between the populations were set based on MFI ($\geq 3000$), area (active $\geq 10\,\mu m^2$, inactive $\geq 5\,\mu m^2$), and roundness (active $\leq 0.83$, inactive $\geq 0.83$) to classify all bile canaliculi as either inactive or active transport.

Lastly, composite images of mitochondrial activity and bile canaliculi were created in the open access image analysis platform, FIJI, using a macro to assign color to TMRM intensity ranges with maximum projections of MFI bile canaliculi images superimposed in 8-bit gray scale[73]. The FIJI plugin, Coloc2, was used to generate Mander's correlations for above zero intensity of channel (M1, M2) and above auto threshold of channel (tM1, tM2) for whole images ($\times 20$, 0.4 NA) and Pearson's correlation, $r$, for selected regions of interest (ROI) for days 2–30. Run parameters were set with a point spread function (PSF) at 3 and Costes randomizations at 100, allowing for minimal influence of noise and independent of image brightness[36,37].

**_P. falciparum_ mosquito infections from in vitro culture**. _P. falciparum_ (strain NF54) was maintained according to standard methods at 37 °C (5% $O_2$ and 5% $CO_2$, nitrogen balanced) in 5% hematocrit (O + blood), 10% AB human serum (The Interstate Blood Bank INC, Memphis, TN, USA), RPMI 1640 medium (Cat No. CUS-0645, KD medical, Columbia, MD, USA), and 2.5% sodium bicarbonate[74]. Two gametocyte cultures were started three days apart using standard methods and an aliquot of each was mixed together when stage V gametocytes were prevalent (day 14–17 of gametocyte cultures) and used to infect day 4–6 old mosquitoes. Infections of laboratory-reared _Anopheles_ (_An._) _stephensi_ (Nijmegen strain) were accomplished using a temperature-controlled Hemotek membrane-feeding device (Discovery Workshops, Accrington, UK) and kept at 80% humidity in 26 °C environmental chambers with 10% sucrose supplemented water[75]. Additionally, _P. falciparum_ (strains NF54-WT and NF54-GFP) infected _An. stephensi_ mosquitoes were shipped live from John Hopkins University Malaria Research Institute (JHUMRI) insectary and parasitology core (Baltimore, MD, USA) on day 10 post-infection and maintained at 80% humidity in 26 °C environmental chambers with 10% sucrose supplemented water. Salivary glands of mosquitoes 14–16 days after infection were aseptically dissected and collected into Schneider's media (Cat No. S9895, Sigma-Aldrich, St. Louis, MO, USA)[76].

**Ethical statement**. The Human subjects protocols for this study was approved by the Institutional Ethics Committee of the Thai Ministry of Public Health, the Human Subjects Research Review Board of the U.S. Army (WRAIR#1949), Ethical Review Committee of Faculty of Tropical Medicine, Mahidol University (TMEC 11-008 and 14–016), the Oxford Tropical Medicine Ethical Committee, Oxford University, England (OxTREC 17-11 and 40–14), and the Cambodian National Ethics Committee for Health Research (101NECHR). The protocols conformed to the Helsinki Declaration on ethical principles for medical research involving human subjects (version 2002) and informed written consent was obtained for all volunteers.

**_P. vivax_ mosquito infections from infected patients**. Isolates of _P. vivax_ were collected from symptomatic patients enrolled in the study after giving written informed consent at the Thai malaria clinic (SMRU) in Tak province, Mae Sod Malaria Clinic (Thailand Ministry of Public Health) in Tak province, and through recruitment via the village malaria workers network and the local health centers in the Mundolkiri Province in Cambodia. A sample of _P. vivax_-infected blood was drawn by venipuncture into heparin tubes. Thick and thin Giemsa-stained smears were prepared after removal of white blood cells (WBCs). A diagnostic PCR was used to confirm microscopic identification that _P. vivax_ was the only _Plasmodium_ species present in the samples used for the study[77].

The mosquito infections for _P. vivax_ were performed using a colony of _An. dirus_ (Bangkok strain), _An. cracens_[78] or _An. dirus_ B in Cambodia. Briefly, 150 µL of RBC pellet from infected patient blood samples was suspended in pooled normal AB human serum to 50% hematocrit. Next, 300 µL of the suspension was fed for 30 min to 5–7 days old female mosquitoes via an artificial membrane attached to a water-jacketed glass feeder maintained at 37 °C. Engorged mosquitoes were kept on 10% sucrose solution and maintained at 26 °C and 80% humidity until dissected. Salivary glands of mosquitoes 14–16 days after infection were dissected and collected as described above. Studies including freshly dissected _P. vivax_ sporozoites were performed at Armed Forces Research Institute of Medical Sciences, Bangkok, Thailand, Shoklo Malaria Research Unit, Mae Sot, Thailand, and Institut Pasteur du Cambodge, Phnom Penh, Cambodia. For some LS studies, mosquitoes carrying _P. vivax_ oocysts were shipped, still pre-infectious, from AFRIMS to the University of South Florida following permit approval by the Thai Ministry of Health, US Center for Disease Control, US Department of Agriculture, and Florida Department of Agriculture.

**Infection with cryopreserved _Plasmodium_ sporozoites**. Previously cryopreserved lots of _P. vivax_ and _P. falciparum_ sporozoites were used to assess PHH invasion and development. Cryopreservation of sporozoites were performed following previously published protocols and thawed at room temperature with dilution in HCM[51,52]. A lot of cryopreserved _P. vivax_ sporozoites refers to a single patient derived case while a lot of cryopreserved _P. falciparum_ sporozoites refers to lab-adapted strain NF54. A cryopreserved _P. vivax_ sporozoite lot was used and

infected at $2.0 \times 10^4$ sporozoites per well (Supplementary Fig. 4a, b, Supplementary Fig. 9, and Supplementary Table 1) or at $1.0 \times 10^4$ sporozoites per well (Fig. 3a and Supplementary Table 3). A cryopreserved _P. falciparum_ sporozoite lot was used and infected at $2.5 \times 10^4$ sporozoites per well (Fig. 3a and Supplementary Table 3).

**Inhibition of Liver Stage Development Assays (ILSDAs)**. After salivary gland dissections at WRAIR, SMRU, IPC, or USF, sporozoites were counted using a hemocytometer and diluted accordingly in HCM. For antibody and serum exposure, sporozoites were incubated with test serum samples in a serially diluted manner (1:2, 1:4, 1:8, 1:16, 1:32, 1:64, 1:128, 1:256 in HCM) or with monoclonal antibodies (80, 40, 20, 10, 5, 2.5, 1.25, 0.25 µg ml$^{-1}$ in HCM) (_P. falciparum_) and 250, 25, 2.5, 0.25 µg ml$^{-1}$ (_P. vivax_) for 20 min at room temperature or overnight at 37 °C, 5% $CO_2$. For experimental control, sporozoites were exposed to pre-bleed mouse sera as well as in absence of an exposure (HCM only) then held at room temperature for 20 min or overnight at 37 °C, 5% $CO_2$. After incubation, $1.0 \times 10^4$ or $1.8 \times 10^4$ sporozoites from each condition were added to wells in duplicates or triplicates, spun down at $200 \times g$ for 5 min, and allowed to invade for 24 h at 37 °C before washing with HCM. Media was changed every 2 days using HCM supplemented with antimicrobial mix until fixation 6 days post initiation.

**_P. vivax_ and _P. falciparum_ antimalarial drug studies**. As stated previously, salivary glands were disrupted and sporozoites were counted on a hemocytometer, diluted in HCM, and $0.5 \times 10^4$ added per well prior to centrifugation at $200 \times g$ for 5 min. Drug regimen was defined by treatment mode, prophylactic or radical cure, with addition of drug on assigned days and HCM supplemented with antimicrobials replenished daily. Compounds were then dissolved in 100% DSMO to stock concentrations of 10 mM and then dispensed and serially diluted in absorption resistant 384-well master plates (Axygen, Tewksbury, MA, USA) using a multichannel pipette. Plated compounds were stored at −80 °C before and after shipment to an endemic laboratory (SMRU in Thailand and IPC in Cambodia) on dry ice. Plated compounds were thawed at room temperature, spun down at $200 \times g$ for 5 min and stored covered with foil tape in an ambient desiccation chamber between daily transfers. Transfer of compounds from the master plate to the 384-well assay microplate was performed using a sterilized, custom made, manual pin tool (VP Scientific, San Diego, CA, USA) engineered to dispense 40 nL of liquid per pin into 40 µL HCM volume in the well to achieve a 1000-fold dilution. Before and after compound transfer the pin tool was cleaned by two wash cycles (dip and blot) in each of 50% DMSO in water, then 70% EtOH in water, then VP clean solution, then water, and finally 100% MeOH, per manufacturer's protocols. Initially, compounds were tested in independent experiments using an 8-point concentration format with 3-fold dilutions (final concentrations of 10 µM to 5 nM). For Single Point assays, compounds were tested and the most potent 24 hits were selected for confirmation and EC$_{50}$ determination in the same 8-point concentration format described above. For each assay plate, both a positive (KDU691) and negative (0.1% DMSO) control were added and a minimum of 8 wells per plate were analyzed. Based on controls, a Z' factor was calculated for each plate and positive value were considered acceptable for data analysis[79].

**Automated image acquisition and data analysis**. At 5−8 days post infection, wells were fixed with 4% paraformaldehyde (PFA) for 10 min at room temperature and washed twice with 1×PBS. For _P. falciparum_ experiments, wells were incubated in blocking buffer (0.03% TritonX-100, 1% (w/v) BSA in 1 × PBS) with mouse anti-GADPH (3.2 µg ml$^{-1}$, 1:50,000 dilution) overnight at 4 °C. For _P. vivax_ experiments, wells were incubated in blocking buffer with polyclonal rabbit or mouse recombinant anti-UIS4 mAB (1 µg ml$^{-1}$, 1:5000 or 1:25,000-fold dilution) overnight at 4 °C. After the wells were washed thrice with 1× PBS and incubated for 1 h at room temperature with goat anti-rabbit or goat anti-mouse Alexa Fluor® 488 conjugate (2 µg ml$^{-1}$, 1:1000-fold dilution; Cat No. A11001, Molecular Probes, ThermoFisher, Waltham, MA, USA) secondary antibody and Hoechst (10 µg ml$^{-1}$, 1:1000-fold dilution) then washed thrice, and filled with 1× PBS for imaging and storage.

Imaging and data analysis of the ILSDAs and drug plates were completed using the Operetta Imaging System and Harmony software 4.1 (Perkin Elmer, Waltham, MA, USA) or ImageXpress and MetaXpress software (Molecular Devices, Sunnyvale, CA, USA). Images were acquired using FITC, DAPI, and brightfield channels at ×20 magnification meaning each well of a 384-well plate is 35 fields of view, tiled together. After image acquisition, hepatocyte nuclei were counted with the DAPI channel and objects greater than 30 µm$^2$ were defined by the software. Parasites were counted with the FITC channel and were identified by area, mean intensity, maximum intensity and cell roundness. To improve the accuracy of the software identifying the parasites, parasites were separated into 2 populations: small and large forms with the large forms divided into 5 subpopulations. The parameters and thresholds used to count the parasites are summarized in Table 1. After the parasites were counted, the five subpopulations of large forms were added together for _P. vivax_ and the small and large forms were added for _P. falciparum_. Drug EC$_{50}$ curves and % inhibition were generated using total population counts and parasite size where controls were calculated as the average of replicates (when performed) using the Levenberg-Marquardt algorithm, using KDU691 as the normalization control, as defined in CDD Vault (Burlingame, CA, USA). Similarly,

$IC_{50}$ curves and % inhibition of ILSDAs was performed using dose-response modeling in Prism (Graphpad, La Jolla, CA, USA) where data were normalized to the negative control (infected well, ICtrl) described by the equation below:

$$\% \text{ Inhibition} = 100 - \left[ \frac{X}{\text{ICtrl}} \times 100 \right]$$

where $X$ is the measured inhibition by the antibody or immune sera and ICtrl is calculated using the average of 12 experimental replicates in the same 384-well plate.

**High-resolution imaging and blood breakthrough assays**. Briefly, the same immunofluorescence assay (IFA) protocol was followed above with the following fold dilutions of monoclonal antibodies obtained from The European Malaria Reagent Repository: mouse anti-GAPDH (Cat No. 7.2) at 1:50,000, mouse anti-EXP1and EXP2 (Cat No. 5.1 and 7.7) at 1:1000, mouse anti-GLURP (clones 22G6, 8B12, 2C7) at 1:200, anti-MSP1 (Cat No. 12.10) at 1:200, and anti-MSP2 (Cat No. 12.3) at 1:200[39–41]. The following antibodies were used; polyclonal rabbit anti-UIS-4 at 1:5000, recombinant mouse UIS-4 at 1:25,000, polyclonal rabbit anti-MIF 1:1000, and monoclonal mouse ACP at 1:1000 and were obtained from Center for Infectious Disease Reasearch, WA, USA[31]. The anti-UIS4 recombinant monoclonal antibody used in this study was derived from a hybridoma expressing anti-UIS4 monoclonal antibody developed by methods previously described[31,80]. We produced recombinant mouse anti-Pvs16 then conjugated to Alexafluor® 647 and used experimental dilutions at 1:50 (Cat No. A20186, Molecular Probes, ThermoFisher, Waltham, MA, USA). We also produced a mouse anti-HSP70 hybridoma supernatant (clone 4C9) and used at 1:100. After primary staining, the wells were washed thrice with 1× PBS and incubated for 1 h at room temperature with either goat anti-rabbit Alexa Fluor® 488 conjugate, goat anti-mouse Alexa Fluor® 488 conjugate, goat anti-rabbit Alexa Fluor® 568 conjugate, or goat anti-mouse Alexa Fluor® 568 conjugate (2 µg ml$^{-1}$, 1:1000-fold dilution; Cat No. A11001, A11034, A11004, A11011, Molecular Probes, ThermoFisher, Waltham, MA, USA) secondary antibody and Hoechst (10 µg ml$^{-1}$, 1:1000-fold dilution) then washed thrice, and filled with 1× PBS for imaging and storage. High-resolution, z-stacked images of P. vivax and P. falciparum LS parasites were captured with a ×100 oil objective, 1.4 NA on a DeltaVision Core system (GE Healthcare Life Sciences, Piscataway Township, NJ, USA). Images were deconvoluted using the softWoRx® image analysis package (GE Healthcare Life Sciences, Piscataway Township, NJ, USA) and z-stacks with added scale bars were processed in FIJI[73]. Images captured with secondary antibody Alexa Fluor® 568 were re-colored to magenta using FIJI in order to meet publication image standards.

*Plasmodium* blood breakthrough studies were performed using either fresh reticulocytes prepared from Duffy antigen positive buffy packs (P. vivax) or fresh O + whole blood (The Interstate Blood Bank INC, Memphis, TN, USA). O$^+$ blood (25 ml) was centrifuged at 2000 RCF for 5 min to remove buffy coat and RBCs were washed thrice with incomplete (no serum or sodium bicarbonate) RPMI 1640 medium (Cat No. CUS-0645, KD medical, Columbia, MD, USA). Similarly, buffy pack blood (60–70 ml) was washed with McCoy's 5A incomplete medium (Cat No. M9309, Sigma-Aldrich, St. Louis, MO, USA), filtered on a NEO1 Leukocyte Reduction filter (Cat No. BPF4, Haemonetics, Braintree, MA, USA) then subjected to a CD71$^+$ immuno-magnetic purification following manufacturer's specifications (Miltenyi Biotech, Bergisch Gladbach, Germany). CD71$^+$ selected cells were washed in McCoy's 5A incomplete medium and New Methylene Blue thin blood smears were performed to check for purity, yielding normally >95%. Reticulocytes and RBCs were kept for up to 7 days at 4 °C at 50% haematocrit in incomplete medium before overlay on mature LS cultures; days 9–12 for P. vivax and days 6–8 for P. falciparum. Prior to addition to LS cultures, reticulocytes or RBCs were pelleted and resuspended in serum containing media then cultures were co-incubated overnight at 37 °C, 5% CO$_2$ after pipette disruption. Smears were collected daily and fixed with methanol or Giemsa stained.

**Statistical analysis**. All graph bars are represented by means with standard deviation (s.d.). Significance of mean was assessed by one-way ANOVA with Dunnett's multiple comparisons (Fig. 2, Fig. 6e, Supplementary Figure 4a, b), one-way ANOVA (nonparametric) followed by Dunn's multiple comparison (Supplementary Figure 4c, d), two-way ANOVA with Dunnett's multiple comparisons (Fig. 3a, b, Supplementary Figure 4), two-way ANOVA with Tukey's multiple comparisons (Fig. 6b) When standard deviation is reported, $n$ represents biological replicates where $n'$ represents experimental replicates per biological replicate.

**Data availability**. The data that support the findings of this study are available from the corresponding author upon request.

## Disclaimer
Material has been reviewed by the Walter Reed Army Institute of Research. There is no objection to its presentation and/or publication. The opinions or assertions contained herein are the private views of the authors, and are not to be construed as official, or as reflecting true views of the Department of the Army or the Department of Defense. The investigators have adhered to the policies for protection of human subjects, as prescribed in AR 70–25.

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

## Acknowledgements

We thank the *P. vivax* patients and residents of Tak Province, Thailand and Mondulkiri Province, Cambodia for their generosity for participation in the study. We thank AFRIMS, SMRU, and IPC insectary department and staff for *An. dirus* mosquito breeding, feeding of *P. vivax* infected blood, and maintenance. Shoklo Malaria Research Unit is part of the Mahidol Oxford University Research Unit, supported by the Wellcome Trust of Great Britain (F.N.). We acknowledge John Hopkins University Malaria Research Institute (JHUMRI) insectary and parasitology core, with special thanks to A.

Tripathi and P. Sinnis for facilitating shipments of *P. falciparum*-infected mosquitoes (J.H.A.). The following reagent obtained through BEI Resources, NIAID, NIH: Monoclonal Antibody 2A10 Anti-*P. falciparum* Circumsporozoite Protein (CSP), MRA-183A, Hybridoma 2F2 Anti-*P. vivax* CSP, MRA-184, and Hybridoma 2E10.E9 Anti-*P. vivax* CSP, MRA-185, contributed by E. Nardin. Monoclonal antibodies 12.10 (anti-MSP1), 5.1 (anti-EXP1), 7.7 (anti-EXP2), clones 22G6, 8B12, 2C7 (anti-GLURP), and 7.2 (anti-GAPDH) were obtained from The European Malaria Reagent Repository (http://www.malariaresearch.eu). HCI data from drug studies was produced in collaboration with the Biomedical Microscopy Core at the University of Georgia (D.E.K.). Funding support was provided by Bill and Melinda Gates Foundation (OPP1023643 to J.H.A., OPP1023601 to D.E.K), Medicines for Malaria Venture (RD/16/1082 and RD/15/022 to D.E.K., RD/2017/0042 to B.W. and A.V.), the Georgia Research Alliance (D.E.K.), and National Institutes of Health (R01AI064478, BAA-NIAID-DAIT-NIHAI2013164 to J.H.A.).

## Author contributions

Overall study design and development: A.R., S.P.M., D.E.K., J.H.A.; *P. vivax* vaccinations and sera: D.E.L., S.A.K.; Coordinating anti-malarial drug assay design: B.C, C.W.M., M.R., M.A.B., S.P.M., D.E.K; UIS4, MIF, ACP antibodies: S.A.M., N.S, S.H.I.K.; *P. vivax* sample collection, mosquito and sporozoite production: R.U., V.C., C.A., A.V, S.D., F.N., B.W.; Sporozoite isolation protocols: A.R., J.H.A.; Cryopreservation protocol: N.S., A.R., S.P.M., S.J.B.; Sporozoite isolation and hepatocyte functional assays: A.R., S.P.M.; Experimental LS assays: A.R., S.P.M.; LS parasite maturation assay: A.R., S.R.A., R.T.L.; Immunofluorescence assay and HCI: A.R., S.P.M., A.J.C., C.C.; High-resolution imaging: A.R. Data analysis and interpretation: A.R., S.P.M., A.J.C., R.H.Y.J., D.E.K., J.H.A.; Manuscript first draft: A.R.; Manuscript additional preparation: A.R., S.P.M., A.J.C., R.H.Y.J., D.E.K., J.H.A. All authors contributed to final, submitted manuscript.

## Additional information

**Competing interests:** The authors declare no competing interests.

