## [Peer Review File · Nature Communications]

Reviewers' comments:

Reviewer #1 (Remarks to the Author):

This is a very nice MS showing robust evidence of a high throughput system to screen for LS antimalarials in both *P. vivax* and *P. falciparum*.

Some comments and concerns down here:

'Unfortunately, 8-aminoquinolones induce toxicity in individuals with glucose-6-phosphate dehydrogenase (G6PD)-deficiencies and suffer altered metabolism with G6PD polymorphisms,' I believe the authors mean CYP polymorphisms instead.

Any information on the CYP background of these PHH used in the experiments? Any speculation on the cause of differences seen on LS parasite development according to lots? Does that explain PQ and TQ responses seen in the study? Keep in mind that genotypes do not necessarily correspond to phenotypes of CYP expression.

Why *P. falciparum* sporozoites were not produced from mosquitoes fed on clinically-isolated blood meals?

'a portion of *P. vivax* late-stage schizonts stained positive for Pvs16, a gametocyte-specific protein, suggesting some LS merozoites are pre-committed as first generation gametocytes prior to the erythrocytic cycle (Fig. 4d and Supplementary Fig. 6)', I believe this is a brand new finding which requests a separate publication. The information is diluted here.

'no merozoites were observed in *P. falciparum* experiments', why was that so?

Was variants of CSP analyzed in the samples used from diverse endemic areas?

'While relative expression of surface receptors could not be analyzed in this study, additional diagnostic tools may be needed to identify biomarkers linked to a compatible donor phenotype.': this is very important information lacking in the paper.

'This mechanism, if confirmed, would help explain vivax transmission between an asymptomatic individual and the mosquito vector': too soon to make this statement, commitment to gametocytogenesis does not mean infectivity to the mosquito.

Not sure if *P. vivax*-infected mosquitos were sent to the US for SG dissection. This is not clearly stated in the MS.

Reviewer #2 (Remarks to the Author):

This is a well-written study of high importance, especially for scientists working on the liver stage of *P. falciparum* and *P. vivax* malaria – two most virulent species affecting half of the Earth's population. Development of an enhanced efficiency and throughput in vitro functional system for fundamental liver stage studies and for screening of pre-erythrocytic vaccines/drugs is highly needed. This is especially important for *P. vivax* studies with its hypnozoite state, which is extremely difficult to study. The manuscript clearly describes how to prepare and use this system, which is more efficient than other currently available approaches to perform this kinds of studies, and meticulously validates all steps in LS development, from sporozoite isolation to liver infection to release of merozoites, and parameters necessary to establish such a functional system. The results presented here not only describe development of an important technical tool, but also provide several interesting findings from the basic point of view on biology/physiology of malaria parasites (e.g. formation of uncharacterized vacuoles, gamete-forming merozoites), and will be of interest to a wide research audience. I have only a few minor notes and suggestions to improve this manuscript, described below.

Line 171-172 and Fig. 3a legend: Donor H is also susceptible to a single species (*P.f.*), but is not mentioned.

Line 195: It is better to reference Fig 3 c and f, not only 3f.

Figure 3 f and g legend: it is better to indicate clearly the days of development in the figure legend for these panels. Line 218: refer to Fig. 4c

Line 220: Better to refer to MSP-1 image in Fig 4a

Fig 6a. It is not clear what is "100% inhibition". On which day the measurements that are shown in the panels were made? It is better when stated not only in the text but also in the Figure legend.

Lines 284-287 and reference to Fig 6d. Something is mixed up. Text describes anti-PvCSP mAb2F2 tested on *P. vivax* LS parasites, and legend (lines 1127-1130) describes effect of Pf mAb 2A10 on Pf LS parasites: "(d) *P. falciparum* SGSs exposed to anti-*P. falciparum* circumsporozoite protein monoclonal antibody 2A10 (PfCSP mAB 2A10) showed high inhibition rates at all concentrations with a similar decreased growth phenotype at 10 µg/mL compared to no antibody control (e, f)."

Lines 294-297 confirm this reviewer's guess (about mixing things) above, as authors talk about PfCSP mAB 2A10 and also refer to Fig 6d. Text and legend should be corrected to prevent all the confusions. It is better to clearly describe what is what on each panel in Fig 6, and make corresponding corrections in the text referring to this Figure.

Line 307. "SAPN" should be spelled out and referred to the previous publication(s).

Line 309. Is word "immunization with" should be inserted after the word "from"? Same on line 312.

Figure 6 legend, line 1132: "Error bars represent mean with s.d...." This is incorrect language, in several Figure and supplementary figure legends, my suggestion is: "Graph bars represent means with s.d. of biological replicates". Similar corrections suggested for other Figure legends.

Figure 7 and its legend. In the legend, atovaquone and PI4K inhibitor KDU691 called control compounds. Thus, it is not clear what is shown on axis Y, which states "inhibition (% control)". Is it inhibition with specific concentration of KDU691? If so, this should be stated in the legend to prevent confusion.

Line 345: it is better to add "last panel" when referring to Fig.7d for clarity. It is also good to refer to Fig.7d after "salinomycin and lasalocid-A" (Fig 7d) on line 247 for clarity.

Supplementary Figure 2. Some numbers are enhanced by bold font, but no explanation why they are enhanced

Supplementary Figure 4 c and d. Graphs with Boxes and whiskers do not explain what boxes and whiskers are.

Supplementary Fig. 7. The Figure legend is not clearly written. The title is "Characterization of effects on *P. vivax* salivary gland sporozoite (PvSGS) invasion with coupled exposure to active complement". Where is active complement in panels a, b, c? What is HCM in panels a and b? It should be spelled. Panel d : "PvSGS exposed to the control serum (pre-bleed) showed moderate inhibition in comparison to the no sera exposure controls". However, the graph shows "infected control" – this is a little confusing and takes time to think what it "infected control". It is better to mark "infected hepatocytes without serum" or simply "no serum".

Andrew V. Oleinikov

Reviewer #3 (Remarks to the Author):

Preclinically testing of the efficacy of candidate therapies against liver stages of Pf and Pv remains challenging due to the scarcity of adequate experimental models. Study of liver stages of parasites causing malaria in humans remains challenging due to their highly restricted species and cellular tropism. Human hepatoma cells (e.g. HC04) have been used to study Pf and Pv infection in vitro but their utility remains limited because of their transformed nature, aberrant cellular signaling and incomplete support of liver stage development. Humanized mice, i.e. mice growing a partially human liver, remain a gold standard and have been shown to faithfully support complete liver stage development of Pf,

Pv and Po. However, generation of human liver chimeric mice is costly, low in throughput, and requires significant technical skills which poses some challenges. It was previously demonstrated that cultures of primary human hepatocytes (PHHs) or stem cell derived hepatocyte like cells (HLCs), especially when plated in co-culture with murine non-parenchymal stromal cells in micropatterned formats, support Pf and Pv LS development. However, infection efficiency was low and parasite development distinct from what has been described in in vivo models, such as liver chimeric mice.

Here, Roth, Maher and colleagues characterized Pf and Pv infection in primary human hepatocyte cultures. Importantly, infections could be downscaled to 384 well plate formats which opens opportunities for conducting primary screens aimed at identifying novel compounds with antimalarial activity. Surprisingly PHHs from some select donors retained some functional activity over 30 days. Relevant to the initial rationale for this study the authors demonstrate that specific PHH donors support complete Pf and Pv liver stage development as evidenced by the size and shape, acquisition of characteristic phenotype markers and release of merozoites capable of infecting reticulocytes that had been overlaid on top of the monocultures. Employing a high content imaging platform the authors provide proof-of-concept for the utility of their PHH platform for anti-parasitic drug and vaccine testing. Overall, this is a well conducted study provided POC for the utility of PHH in microscale formats for antimalarial drug development. While this platform may not be perfect it certainly represents an important new high through-put technology holding potential to prioritize antimalarial drug candidates. However, the authors should be more explicit about the limitations of the platform throughout the manuscript (as specified below).

1. It is a major shortcoming of the study and a considerable challenge in the field that the susceptibility of different hepatocyte donors to Pf and Pv sporozoites differs significantly. This will arguably limit the utility of the platform and increase the costs as specific lots need to be pre-screened of which then only limited quantities of PHHs may be available. This should be critically discussed in the paper and also highlighted in the abstract. Also, what is unique about the donors that appeared to be more susceptible to infection? The authors mention in the legend for figure S1: "PHH donor lots from Bioreclamation IVT (BIVT) along with other commercially available lots were screened using previously cryopreserved P. vivax sporozoites showing significant reduction in sporozoite invasion and LS development in non-BIVT lots. " Rather than categorizing the susceptibility by supplier the authors should make an effort to correlate donor characteristics to susceptibility. The authors speculate in the discussion that differences in CD81 and SRB1 expression or inadequate cryopreservation may account for this phenotype but there are likely many other parameters that affect this process, including but not limited to other cell intrinsic factors, medication of the organ donor prior to tissue harvest, perfusion stress...Having a defined set of biomarkers correlating with susceptibility to Pf and/or Pv infection would certainly be of great utility to identify rapidly suitable donor lots without having to rely on costly and labor-intensive functional testing.

2. Along the same lines it is concerning that in some donors appear to be only susceptible to Pf or Pv arguing that the normal, highly differentiated hepatic phenotype is not well recapitulated in this specific culture format.

3. It is unclear why/whether simply plating PHHs in a smaller culture format (here 384 well

plates) changes the physiology of the cells in a way that they become (more?) susceptible to Pf and Pv infection. A stand-alone statement such as "We discovered that the small-scale collagen-treated surface area of particular commercially available 384-well plates coupled with our methodology provides a suitable microphysiological environment for long-term cultivation of PHH" (page 6/7, line 120)" does not provide information how the authors reached that conclusion. The authors should perform back-to-back comparisons in other larger scale formats (at least 24 and 96 well plates) to provide experimental evidence for whether the plating format is responsible for the observed phenotype.

4. It is surprising that cultures with specific hepatocyte lots appear stable for over 30 days which stands in stark contrast to previously published work (e.g. Khetani & Bhatia (2007) Nature Biotechnology). The authors provide some but overall very limited data on the functional activity of the hepatocyte cultures (Fig. 2b and Fig S1). The authors should also clearly specify which donors were subjected to these tests and how reproducible those were. This is critical to support the claim to the general utility of the platform.

5. Line 243/244: How exactly do the authors define and distinguish between "immature" and "mature" hypnozoites? Are there clear maturation markers that change between days 4-5 (early treatment) and beyond d5? Day 5 seems to be a rather early delineation point.

6. Line 291: This reviewer agrees that the quantitative HCI analysis anti Pf and Pv antibody blocking experiments yielded some intriguing results. However, highlighting that the "platform is suitable for assessing not only antibody efficacy, but also a better understanding of immune antibody mechanisms of action" is an overstatement. Antibody effector functions going beyond neutralization/opsonization (complement activation, phagocytosis etc.) are not being modeled in this platform.

7. Please provide detailed protocols on the Pv cryopreservation procedure. How efficient was the cryo recovery?

8. The drug treatment dosing analysis are very interesting. Have the authors validated the antimalarial activity of at least those compounds that appear effective in one hepatocyte donors across others?

9. It is peculiar that dosing with primaquine appeared to have no effect established liver stages in this model. This stands in contrast to observations by Mikolajczak et al. (2015 CH&M) demonstrating efficacy in a humanized liver mouse model.

Minor comments:

10. Page 7, line 132: Albumin secretion is technically not a metric for metabolic activity

11. Supplementary table 1: it is unclear what the authors exactly mean by "Fibroblasts after 2 weeks". It seems that PHHs from these specific donors rapidly de-differentiated and lost their characteristic hepatocyte morphology but they certainly did not convert into other cell types.

12. "genetically-modified HepRG hepatocyte line" (line 167). Do the authors mean HepaRG cells? HepaRG cells have been described as a hepatic progenitor cell line not a hepatocyte line. Please define the source how these cells, how they have been treated to differentiate them and how have been genetically modified.

13. Line 251: There is a reference to a submitted MS. Additional information should be provided about the anti-UIS4 antibody in the present study.

14. Line 343: typo "8-aminoquinolnes"

Responses to major concerns

We agree that these two aspects, i.e. donor variability and characterization of 384 format effect on hepatocytes, are important. For others to perform similar studies with our platform, routine availability of multiple different donor lots will not be an obstacle. The commercial source of cryopreserved primary human hepatocytes (PHH), Bioreclamation IVT (<http://www.bioreclamationivt.com/>), provides multiple free samples for screening. We have additional information on lots screened and their detailed characteristics in Supp Table 2; CYP activity is openly accessible on the BIVT web site.

To address concerns over lot-specific effects on drug studies, we added data (Supp Fig 10) to include drugs tested in multiple hepatocyte lots to demonstrate consistency of drug susceptibility data. A section of discussion related to variability has also been added (lines 579-590).

To specifically address the effect of 384 well format on hepatocyte stability, we evaluated different well formats for the ability to support long-term viability of PHH with in vivo-like primary cell phenotypes. These comparison data have been added to Supp Fig 1. As these primary culture results demonstrate, when these same hepatocytes are seeded into larger well formats, the cells will die within one week (consistent with the literature data). These results show that the small culture area of a 384-well plate is superior to larger well formats in maintaining PHH viability and primary hepatocyte properties. The exact mechanism of this phenomenon is the subject of our currently ongoing research.

Reviewer #1 (Remarks to the Author):

This is a very nice MS showing robust evidence of a high throughput system to screen for LS antimalarials in both *P. vivax* and *P. falciparum*.

We thank the reviewer for these comments.

Some comments and concerns down here:

‘Unfortunately, 8-aminoquinolones induce toxicity in individuals with glucose-6-phosphate dehydrogenase (G6PD)-deficiencies and suffer altered metabolism with G6PD polymorphisms,’ I believe the authors mean CYP polymorphisms instead.

This sentence has been revised.

Revised sentence: “Unfortunately, use of 8-aminoquinolones is contraindicated in many malaria endemic countries because of its toxicity in individuals with some glucose-6-phosphate dehydrogenase (G6PD) polymorphisms, restricting mass drug administration campaigns in regions where high-risk favisms are common¹⁷.”

Any information on the CYP background of these PHH used in the experiments? Any speculation on the cause of differences seen on LS parasite development according to lots? Does that explain PQ and TQ responses seen in the study? Keep in mind that genotypes do not necessarily correspond to phenotypes of CYP expression.

As indicated above, we added Supplemental Table 2, containing this information, based on detailed characteristics of each PHH lot available from Bioreclamation IVT.

As far as LS parasite development, we agree that lot-specific parasite biology is not unexpected; these are clonal lots of human hepatocytes infected with human patient isolated-sporozoites. We likely are seeing some natural variation in the population of human donors. However, measuring this population variation would be confounded by the possible effects of cryopreservation, which likely magnify lot variation, making it less informative to characterize.

A discussion section has been added (lines 579-590) addressing differences in CYP metabolism in drug assays and explaining that PQ activity is unique in that it takes weeks for treated parasites to clear from the host cell. This was first described in 1983 (Y. Boulard, I. Landau, F. Miltgen, D. S. Ellis, W. Peters, The chemotherapy of rodent malaria, XXXIV. Causal prophylaxis Part III: Ultrastructural changes induced in exo-erythrocytic schizonts of *Plasmodium yoelii yoelii* by primaquine. *Ann Trop Med Parasitol* **77**, 555-568 (1983). We have performed assessments of several drugs in different donors and we observed minor donor-to-donor variation in drug outcome for non-prodrugs; this has been added as Supp Fig10. Although we know little about how they work, many consider PQ and TQ as ideal positive controls for liver stage activity based on clinical evidences. Unfortunately, there are few data characterizing activity in vitro and their pharmacodynamic effects on parasites in vivo are extremely complicated. Additionally, not all treatments achieve radical cure and both PQ and TQ can be toxic. Finally, the drugs are extensively metabolized and the exact metabolite species producing anti-parasitic activity remain unknown. These complications plus the current data from our in vitro model and from humanized mouse models suggest slow clearance of affected parasites and that PQ and TQ are of limited use as controls in these assays.

Why *P. falciparum* sporozoites were not produced from mosquitoes fed on clinically-isolated blood meals?

Sporozoites are a critical and limiting reagent in our study. The use of laboratory-reared *Anopheles* infected with a culture-adapted line of *P. falciparum* helps control some of the variables that occur when using field isolates. In addition, the cost would be about 5-fold or higher with producing *P. falciparum* sporozoites from clinical isolates for this study.

For *P. vivax* sporozoites, we had no choice, as continuous in vitro production of gametocytes has not been demonstrated. Furthermore, our collaborators were located in SE Asia, where *falciparum* malaria transmission is remarkably low, and our work in SE Asia has been funded to specifically assay *P. vivax*.

'a portion of *P. vivax* late-stage schizonts stained positive for Pvs16, a gametocyte-specific protein, suggesting some LS merozoites are pre-committed as first generation gametocytes prior to the erythrocytic cycle (Fig. 4d and Supplementary Fig. 6)', I believe this is a brand new finding which requests a separate publication. The information is diluted here.

We agree that this is a very exciting discovery and explains the well-known phenomenon for vivax malaria of mosquito transmission occurring before the onset of clinical symptoms. Considering its significance and impact, we consider it important to include this important discovery. We also note that another reviewer found this piece of data added value to the manuscript.

'no merozoites were observed in *P. falciparum* experiments', why was that so?

We think there might be a technical reason for this. Although *P. falciparum* merozoites were reported to occur in an *in vivo* humanized liver mouse model, merozoites were not observed in a different *in vitro* model (March et al. 2013. Cell Host & Microbe 14, 104–115). We speculate that *P. falciparum* merozoites exist for shorter time, since thin blood smears were made from our LS cultures at only 2 collection points due to limited resources (in comparison to 10 for *P. vivax* samples) and only infected RBCs were observed. We could have missed their formation. Please keep in mind the key endpoint from the experiment was successful completion of LS development and invasion of erythrocytes.

Were variants of CSP analyzed in the samples used from diverse endemic areas?

Both types of *P. vivax* CSP variants, VK210 and VK247, were used in these studies, but no obvious differences in parasite development or phenotypic responses were observed.

'While relative expression of surface receptors could not be analyzed in this study, additional diagnostic tools may be needed to identify biomarkers linked to a compatible donor phenotype.': this is very important information lacking in the paper.

We agree that this is an important variable and are working towards identification of relevant biomarkers of infectivity. While RNAseq analysis was conducted on the different PHH lots, there was no obvious biomarker associated with differences in infection rates. Considering the observed variation in expression profiles, this will require much more extensive analysis. Still, it is important to note that the relatively simple indicators presented in Figure 2 provide reliable indicators for overall PHH health (mitochondrial homeostasis) and appropriate primary hepatocyte metabolic activity (bile duct formation and excretion) for this assay

'This mechanism, if confirmed, would help explain vivax transmission between an asymptomatic individual and the mosquito vector': too soon to make this statement, commitment to gametocytogenesis does not mean infectivity to the mosquito.

Please see our response provided above. We agree the idea is a hypothesis at this point, but communicating the possibility of producing gametocytes from liver culture is an incredibly important find to share with the field so efforts are made to experimentally confirm if liver parasites can form infectious gametocytes. If confirmed, this would be a key fundamental

understanding for vivax epidemiology. Confirmation is outside the scope of this manuscript.

Not sure if *P. vivax*-infected mosquitos were sent to the US for SG dissection. This is not clearly stated in the MS.

Yes, infected mosquitoes were sent to the USA and studies were done in SE Asia – the methods were revised accordingly.

Reviewer #2 (Remarks to the Author):

This is a well-written study of high importance, especially for scientists working on the liver stage of *P. falciparum* and *P. vivax* malaria – two most virulent species affecting half of the Earth's population. Development of an enhanced efficiency and throughput in vitro functional system for fundamental liver stage studies and for screening of pre-erythrocytic vaccines/drugs is highly needed. This is especially important for *P. vivax* studies with its hypnozoite state, which is extremely difficult to study. The manuscript clearly describes how to prepare and use this system, which is more efficient than other currently available approaches to perform this kinds of studies, and meticulously validates all steps in LS development, from sporozoite isolation to liver infection to release of merozoites, and parameters necessary to establish such a functional system. The results presented here not only describe development of an important technical tool, but also provide several interesting findings from the basic point of view on biology/physiology of malaria parasites (e.g. formation of uncharacterized vacuoles, gamete-forming merozoites), and will be of interest to a wide research audience. I have only a few minor notes and suggestions to improve this manuscript, described below.

We thank the reviewer for these comments.

Line 171-172 and Fig. 3a legend: Donor H is also susceptible to a single species (*P.f.*), but is not mentioned.

The legend was revised accordingly.

Line 195: It is better to reference Fig 3 c and f, not only 3f.

This was revised accordingly.

Figure 3 f and g legend: it is better to indicate clearly the days of development in the figure legend for these panels. Line 218: refer to Fig. 4c

This was revised accordingly.

Line 220: Better to refer to MSP-1 image in Fig 4a

This was revised accordingly.

Fig 6a. It is not clear what is "100% inhibition". On which day the measurements that are shown in the panels were made? It is better when stated not only in the text but also in the Figure legend.

This was revised accordingly. “100% inhibition” was replaced by “with complete inhibition (100%) of PvSGSs invasion and LS development.” Also, the day (day 6) was added to figure legend.

Lines 284-287 and reference to Fig 6d. Something is mixed up. Text describes anti-PvCSP mAb2F2 tested on *P. vivax* LS parasites, and legend (lines 1127-1130) describes effect of Pf mAb 2A10 on Pf LS parasites: “(d) *P. falciparum* SGSs exposed to anti-*P. falciparum* circumsporozoite protein monoclonal antibody 2A10 (PfCSP mAb 2A10) showed high inhibition rates at all concentrations with a similar decreased growth phenotype at 10 µg/mL compared to no antibody control (e, f).”

This was revised accordingly to correctly reference Fig. 6a.

Lines 294-297 confirm this reviewer’s guess (about mixing things) above, as authors talk about PfCSP mAb 2A10 and also refer to Fig 6d. Text and legend should be corrected to prevent all the confusions. It is better to clearly describe what is what on each panel in Fig 6, and make corresponding corrections in the text referring to this Figure.

This was revised accordingly to correctly references Fig. 6 b, c describing LS size reduction. Fig. 6 legend was adjusted accordingly and modified.

Revised:

Figure 6 | *Plasmodium falciparum* and *P. vivax* inhibition of liver stage development assays (ILSDAs). (a) *P. vivax* salivary gland sporozoites (PvSGSs) exposed to anti-*P. vivax* circumsporozoite protein monoclonal antibody 2F2 (PvCSP mAb 2F2) showed a concentration dependent dose response with complete inhibition (100%) of PvSGSs invasion and LS development at 250 µg/mL (b, c) Day 6 developing Pv liver stage (LS) parasites showed a decreased growth (µm²) phenotype to the 25 µg/mL PvCSP mAb 2F2 concentration compared to the no antibody control. (d) *P. falciparum* SGSs (PfSGSs) exposed to anti-*P. falciparum* circumsporozoite protein monoclonal antibody 2A10 (PfCSP mAb 2A10) showed complete inhibition (100%) at concentrations 80 and 40 µg/mL with > 50% inhibition at remaining concentrations. (e, f) Similarly, day 6 Pf LS parasites had a decreased growth phenotype at the 10 µg/mL PfCSP mAb 2A10 concentration compared to the no antibody control. (g) Dose response ILSDAs for test sera immunized against nanoparticles G2, G3, G6, and G7 (top), and representative images of day 6 Pf LS parasite forms exposed to a 1:16 serum dilution (bottom). Graph bars represent means with s.d. biological replicates (n = 3, (a, b) and n = 2, (d, e)) with experimental replicates (n = 2, (a, b, d, e)). Statistical significance was determined for using two-way ANOVA followed by Tukey’s multiple comparisons to control where values are represented as $P < 0.0001$ (****) and no significance (ns) (b) or was determined using one-way ANOVA followed by Dunnett’s multiple comparisons to control where values are represented by $P < 0.01$ (**) and no significance (ns) (e). Grey scale bars represent 10 µm.

Line 307. “SAPN” should be spelled out and referred to the previous publication(s).

This was revised accordingly.

Line 309. Is word “immunization with” should be inserted after the word “from”? Same on line 312.

This was revised accordingly.

Figure 6 legend, line 1132: “Error bars represent mean with s.d....” This is incorrect language, in several Figure and supplementary figure legends, my suggestion is: “Graph bars represent means with s.d. of biological replicates”. Similar corrections suggested for other Figure legends.

This was revised accordingly. All figure legends were amended stating “Graph bars represent means with s.d” followed by replicate type (experimental and/or biological) with the number of replicates (n). Additionally, if statistical analysis was performed, we specifically listed the method used with representation of the p value reported.

Figure 7 and its legend. In the legend, atovaquone and PI4K inhibitor KDU691 called control compounds. Thus, it is not clear what is shown on axis Y, which states “inhibition (% control)”. Is it inhibition with specific concentration of KDU691? If so, this should be stated in the legend to prevent confusion.

This was revised accordingly. The normalization control was stated as KDU691 in the legend.

Line 345: it is better to add “last panel” when referring to Fig.7d for clarity. It is also good to refer to Fig.7d after “salinomycin and lasalocid-A” (Fig 7d) on line 247 for clarity.

We do not understand the requested change. The sentence refers to several ionophores, all included in Fig7d.

Supplementary Figure 2. Some numbers are enhanced by bold font, but no explanation why they are enhanced

This was revised accordingly. The numbers were un-bolded and changed to regular font.

Supplementary Figure 4 c and d. Graphs with Boxes and whiskers do not explain what boxes and whiskers are.

This was revised accordingly. Supplementary Figure 4 parts c and d were revised to bar graphs (no longer box and whisker) and figure legend was changed to appropriately explain.

Revised:

Supplemental Figure 4 | *P. vivax* susceptibility and phenotypic characterization in primary human hepatocytes (PHHs). (a) PHH donor lots from Bioreclamation IVT (BIVT) along with other commercially available lots were screened using previously cryopreserved *P. vivax* sporozoites showing significant reduction in sporozoite invasion and LS development in non-BIVT lots. (b) Seeding densities ranging from 1.2×10^4 to 2.2×10^4 were tested using two top tier PHH donor lots indicating the seeding density of 1.8×10^4 and infection of the PHH between day 4–5 post seed yield highest *P. vivax* LS parasites from separate cases. (c) PvSGSs harvested on day 16 were inoculated at 5,000 sporozoites/well into PDC PHHs at different days post seed, showing

highest total LS parasites per well on day 4. **(d)** PvSGS were harvested from mosquitoes day 14–16 post ingestion of infected blood meal and inoculated at 5,000 sporozoites/well into day 2 post-seeded PHH. Graph bars represent means with s.d. from experimental replicates ($n = 2$). Statistical significance determined using two-way ANOVA followed by Dunnett's multiple comparisons to PDC **(a)** or 1.8×10^4 day 5 **(b)** where significance is presented by $P < 0.05$ (*) or $P < 0.0001$ (****). Statistical significance determined using one-way ANOVA (nonparametric) followed by Dunn's multiple comparison **(c, d)** where significance is presented by $P < 0.01$ (*) and $P < 0.005$ (**).

Supplementary Fig. 7. The Figure legend is not clearly written. The title is "Characterization of effects on *P. vivax* salivary gland sporozoite (PvSGS) invasion with coupled exposure to active complement". Where is active complement in panels a, b, c? What is HCM in panels a and b? It should be spelled. Panel d : "PvSGS exposed to the control serum (pre-bleed) showed moderate inhibition in comparison to the no sera exposure controls". However, the graph shows "infected control" – this is a little confusing and takes time to think what it "infected control". It is better to mark "infected hepatocytes without serum" or simply "no serum".

This was confusing and was revised accordingly. The missing graph error bars were added and the graph legends were revised based upon reviewers suggestions.

Revised:

Supplementary Figure 7 | Media type effects *P. vivax* salivary gland sporozoites (PvSGSs) viability and invasion. (a) PvSGSs incubated in phosphate buffered saline (PBS) for 20-minute at room temperature with anti-PvCSP mAB 2F2 showed an increased % inhibition compared to PvSGSs incubated in hepatocyte culture media (HCM), however, this likely unviable sporozoites leading to reduced invasion rates. (b) Similarly, PvSGSs incubated in phosphate buffered saline (PBS) for 20-minute at room temperature with a non-species-specific antibody (anti-PfCSP mAB 2A10) showed inhibition when there should not be an inhibitory effect. (c) Comparing 24 hour vs. 20 minute PvSGS exposure to anti-PvCSP mAB 2F2 shows a slight curve shift, however, the data goodness of fit increases. (d) PvSGS exposed to the control rabbit serum (pre-bleed) showed moderate inhibition in comparison to the no rabbit serum controls. Therefore, IC_{50} curves for screened sera samples (FMP014/ALF, FMP014/ALFQ, FMP014V/ALF, and FMP014V/ALFQ) were generated by accounting for existing inhibition from serum. Graph error bars represent mean with s.d. from biological replicates ($n = 3$) and experimental replicates ($n = 2$).

Andrew V. Oleinikov

Reviewer #3 (Remarks to the Author):

Preclinically testing of the efficacy of candidate therapies against liver stages of Pf and Pv remains challenging due to the scarcity of adequate experimental models. Study of liver stages of parasites causing malaria in humans remains challenging due to their highly restricted species and cellular tropism. Human hepatoma cells (e.g. HC04) have been used to study Pf and Pv infection *in vitro* but their utility remains limited because of their transformed nature, aberrant cellular signaling and incomplete support of liver stage development. Humanized mice, i.e. mice growing a partially human liver, remain a gold standard and have been shown to faithfully support complete liver stage development of Pf, Pv and Po. However, generation of human liver chimeric mice is costly, low in throughput, and requires significant technical skills which poses some challenges. It was previously demonstrated that cultures of primary human hepatocytes (PHHs) or stem cell derived hepatocyte like cells (HLCs), especially when plated in co-culture with murine non-parenchymal stromal cells in micropatterned formats, support Pf and Pv LS development. However, infection efficiency was low and parasite development distinct from what has been described in *in vivo* models, such as liver chimeric mice.

Here, Roth, Maher and colleagues characterized Pf and Pv infection in primary human hepatocyte cultures. Importantly, infections could be downscaled to 384 well plate formats which opens opportunities for conducting primary screens aimed at identifying novel compounds with antimalarial activity. Surprising PHHs from some select donors retained some functional activity over 30 days. Relevant to the initial rationale for this study the authors demonstrate that specific PHH donors support complete Pf and Pv liver stage development as evidenced by the size and shape, acquisition of characteristic phenotype markers and release of merozoites capable of infecting reticulocytes that had been overlaid on top of the monocultures. Employing a high content imaging platform the authors provide proof-of-concept for the utility of their PHH platform for anti-parasitic drug and vaccine testing. Overall, this is a well conducted study provided POC for the utility of PHH in microscale formats for antimalarial drug development. While this platform may not be perfect it certainly represents an important new high through-put technology holding potential to prioritize antimalarial drug candidates. However, the authors should be more explicit about the limitations of the platform throughout the manuscript (as specified below).

We thank the reviewer for these comments. In the revised manuscript, the major limitations of the new LS-HCI screening assay are included. However, it is also important to note that there are no important barriers to other malaria researchers adopting the use of this innovative *in vitro* liver platform since both the PHH and 384-well culture dish are 'off-the-shelf' items from commercial vendors.

1. It is a major shortcoming of the study and a considerable challenge in the field that the susceptibility of different hepatocyte donors to Pf and Pv sporozoites differs significantly. This will arguably limit the utility of the platform and increase the costs as specific lots need to be pre-screened of which then only limited quantities of PHHs may be available. This should be critically discussed in the paper and also highlighted in the abstract.

response above under "Responses to major concerns"

Also, what is unique about the donors that appeared to be more susceptible to infection? The authors mention in the legend for figure S1: “PHH donor lots from Bioreclamation IVT (BIVT) along with other commercially available lots were screened using previously cryopreserved *P. vivax* sporozoites showing significant reduction in sporozoite invasion and LS development in non-BIVT lots.” Rather than categorizing the susceptibility by supplier the authors should make an effort to correlate donor characteristics to susceptibility. The authors speculate in the discussion that differences in CD81 and SRB1 expression or inadequate cryopreservation may account for this phenotype but there are likely many other parameters that affect this process, including but not limited to other cell intrinsic factors, medication of the organ donor prior to tissue harvest, perfusion stress...Having a defined set of biomarkers correlating with susceptibility to Pf and/or Pv infection would certainly be of great utility to identify rapidly suitable donor lots without having to rely on costly and labor-intensive functional testing.

As noted in our response to a similar comment by reviewer 1, we agree that this is a topic of great interest and an important topic for future research. However, pre-screening suitable lots of cryopreserved hepatocytes for a desired phenotypes is standard practice generally for *in vitro* liver studies (i.e., metabolism studies by pharma), and because it is a routine procedure it is rarely highlighted in the materials and methods. Already intrinsically complex, analysis of hepatocytes in our *in vitro* culture system is further complicated by the PHH regaining *in vivo*-like primary cell characteristics with 3-dimensional organoid architecture, including formation of bile canaliculi. While we perform studies to identify such a desired biomarker in this complex system, we already have the key phenotype to efficiently determine suitable donor lots by the functional screening for the ability to support infection and development of *P. vivax* and/or *P. falciparum*. Combined with the analysis of basic morphology, mitochondrial activity and active transport, the functional assays proved to be the most dependable indicator of future assay reliability. Finally, considering the relative cost and resource-intensive nature of the downstream work, a functional ‘pre-screen’ assay would be a standard practice. It is also worth noting, that compared to the rodent malaria model systems, there is little that is not costly and labor-intensive about liver stage research with the human malaria parasites.

2. Along the same lines it is concerning that in the some donors appear to be only susceptible to Pf or Pv arguing that the normal, highly differentiated hepatic phenotype is not well recapitulated in this specific culture format.

We think this is an important discovery, and perhaps surprising only because there have been no prior studies comparing side-by-side infection rates of different parasite isolates into human hepatocytes from multiple donors. A recently published study, using a hepatocyte micropatterned co-culture system (Gural et al., 2018, Cell Host & Microbe 23, 1–12), demonstrated similarly variable infection rates with different *P. vivax* isolates, albeit at a much lower infection level than in our study. It is premature to know if the differences in LS outcomes for *P. vivax* and *P. falciparum* are inherent characteristics of donor or parasite.

We find that, more often than not, primary hepatic phenotypes are maintained in long term culture in our format. The hepatic phenotypes for each lot we tested are displayed in Supp Table 3. We agree that not all lots can be maintained, but there has been no assessment by ourselves or published by others to assess if a single type monolayer-based culture system can maintain 100% of lots tested. Indeed, lot stability could be irreversibly destroyed by aberrant cryopreservation. As an indication that the cryopreservation itself is foreboding, for all the lots

produced by BIVT and other companies, each is assessed for attachment to glass/plastic and many are considered 'non-plateable' and sold in suspension. We have not seen a publication in which this attribute can be reversed.

3. It is unclear why/whether simply plating PHHs in a smaller culture formats (here 384 well plates) changes the physiology of the cells in a way that they become (more?) susceptible to Pf and Pv infection. A stand-alone statement such as "We discovered that the small-scale collagen-treated surface area of particular commercially available 384-well plates coupled with our methodology provides a suitable microphysiological environment for long-term cultivation of PHH" (page 6/7, line 120)" does not provide information how the authors reached that conclusion. The authors should perform back-to-back comparisons in other larger scale formats (at least 24 and 96 well plates) to provide experimental evidence for whether the plating format is responsible for the observed phenotype.

We agree, it is surprising that a few simple modifications provide such an amazingly different outcome in PHH physiology. In addition to the collagen-treated 384-well Greiner plate, we provide critical information on establishing the PHH in culture, including cell seeding density and volume along with timing and volume of the first change in cell culture medium.

Simple visual imaging of PHH in the larger cell culture formats demonstrate these platforms are inadequate. We added these data as Supp Fig 1. We assert that the literature, which contains much negative data that PHH cannot be maintained in standard culture formats, is only describing negative data because optimizations like the ones we lay out in this manuscript were not attempted. The added data is shown below:

The above images show long-term assessment of primary human hepatocytes (PHH) from Bioreclamation IVT, Inc. in 12-well, 24-well, and 384-well formats. Images taken in bright field at 20x, 0.4 NA showing days 8 and 28 post seed. At day 8, PHH in 12-well have detached while PHH in 24-well have begun apoptotic degradation and detachment. Only PHH in 384-well have a persistent monolayer with polarized cells and visible tight junctions still present at day 28. Standard seed densities were used with 1×10^6 for 12-wells, 2.5×10^5 for 24-wells and 1.8×10^4 for 384-wells.

4. It is surprising that cultures with specific hepatocyte lots appear stable for over 30 days which stands in stark contrast to previously published work (e.g. Khetani & Bhatia (2007) Nature Biotechnology). The authors provide some but overall very limited data on the functional activity of the hepatocyte cultures (Fig. 2b and Fig S1). The authors should also clearly specify which donors were subjected to these tests and how reproducible those were. This is critical to support the claim to the general utility of the platform.

As indicated above, relatively minor changes in plate, medium, and protocol provided dramatically different outcomes in PHH *in vitro* physiology, including extended longevity. We interpret the other studies slightly differently. In the pioneering report of Khetani & Bhatia, micropatterned primary human hepatocyte co-cultures (MPCCs) was demonstrated to promote improved hepatocyte physiology; in essence, the surrounding fibroblasts provide a confined physical space limiting spreading/migration of cultured hepatocytes thereby allowing primary hepatocyte survival in a culture that would otherwise be detrimental. In the group's most recent study (Garul et al., *ibid*), results with the MPCC system were further improved in the 384-well format versus the 96-well format. Our results suggest that within the confined culture space of a 384-well the technically-challenging micropatterning is not really necessary when other *in vitro* culture parameters (medium, protocol, PHH) are optimized. Initially, sample aliquots of several PHH lots are tested for "plateability" and re-acquisition of healthy physiologic characteristics (form bile canaliculi, develop active transport, establish albumin production, etc...) as described in the manuscript (e.g., Fig. 2b and Fig S1). PHH that successfully establish in culture are then tested for ability to support sporozoite infection and LS development of *P. vivax* and *P. falciparum*. To support the claim of the general utility of the platform, additional information on the lots tested has been included as well as phenotypes that were assessed Supp Table 2 and 3.

5. Line 243/244: How exactly do the authors define and distinguish between "immature" and "mature" hypnozoites? Are there clear maturation markers that change between days 4-5 (early treatment) and beyond d5? Day 5 seems to be a rather early delineation point.

The concept of hypnozoite maturation has been presented in previous studies and most notably in *P. vivax* LS infections in the FRG *in vivo* model (Mikolajczak et al., 2015, Cell Host & Microbe 17, 526–535). Beginning about day 3 post sporozoite infection two distinct size populations of LS parasites can be differentiated. The large LS types grow rapidly in size to become mature schizonts by about day 10, while the small types grow incrementally in size until about day 14 where the size essentially plateaus. Work with *P. cynomolgi* (A. M. Zeeman et al., PI4 Kinase Is a Prophylactic but Not Radical Curative Target in Plasmodium vivax-Type Malaria Parasites. *Antimicrob Agents Chemother* 60, 2858-2863 (2016)) has shown that drugging of hypnozoites before day 5 can result in hypnozoite death but after day 5 is ineffective. These data, confirmed in our assay, suggest that hypnozoite drug targets are still susceptible until the form is completely quiescent, and this process takes several days. The Zeeman publication is referenced in our manuscript to expand on this observation.

6. Line 291: This reviewer agrees that the quantitative HCl analysis anti Pf and Pv antibody blocking experiments yielded some intriguing results. However, highlighting that the "platform is suitable for assessing not only antibody efficacy, but also a better understanding of immune antibody mechanisms of action" is an overstatement. Antibody effector functions going beyond

neutralization/opsonization (complement activation, phagocytosis etc.) are not being modeled in this platform.

This has been amended. The primary basis of our excitement in the antibody inhibition studies is the novel observation of the attenuating effect on intracellular LS development by exposing sporozoites to immune antibody before hepatocyte infection.

Revised:

“While further study is needed, these results show our platform is suitable for gaining a deeper understanding of the activity of anti-CSP antibodies as we identified and quantified inhibitory effects manifesting during and after hepatocyte invasion.”

7. Please provide detailed protocols on the Pv cryopreservation procedure. How efficient was the cryo recovery?

The cryopreservation protocol followed in this study uses a commercially available cryoprotectant, CryoStor CS2, and has been published for use with *P. vivax* and *P. falciparum* sporozoites (Singh et al., Parasitol Int. 2016 Oct;65(5 Pt B):552-557) and *P. berghei* sporozoites (Singh et al., PLoS ONE 12(5): e0177304). *P. berghei* sporozoites cryopreserved in CryoStor CS2 achieved 38% complete development of hepatic stages in HC-04 and 100% mice become infected with a dose of 10,000 sporozoites (smaller doses were not tested).

8. The drug treatment dosing analysis are very interesting. Have the authors validated the antimalarial activity of at least those compounds that appear effective in one hepatocyte donors across others?

The Supp Figure 10 has been added showing dose response to several drugs in 3 different donors. A discussion section has been added discussing these data. In short, for drugs active without metabolism, we see no drastic differences between donors. With metabolized drugs, like primaquine, there are some differences observed as expected from donor metabolism variations.

9. It is peculiar that dosing with primaquine appeared to have no effect established liver stages in this model. This stands in contrast to observations by Mikolajczak et al. (2015 CH&M) demonstrating efficacy in a humanized liver mouse model.

This is correct but the original PQ data referenced (Mikolajczak et al., *ibid*) was difficult to reproduce in multiple labs. Ongoing studies indicate that clearance requires >1 week of experimental time frame. We have added long-term clearance requirements to the discussion.

Minor comments:

10. Page 7, line 132: Albumin secretion is technically not a metric for metabolic activity

This is true, but it is nonetheless commonly reported as an indicator of healthy physiology for cultured hepatocytes. We added the term “functionality.”

11. Supplementary table 1: it is unclear what the authors exactly mean by “Fibroblasts after 2

weeks". It seems that PHHs from these specific donors rapidly de-differentiated and lost their characteristic hepatocyte morphology but they certainly did not convert into other cell types.

The text was revised to clarify that some companies' isolation protocols are inadequate at removing additional cell types, such as fibroblasts, and these contaminate the hepatocyte cell preparations. We were not suggesting that hepatocytes turn into another cell type.

12. "genetically-modified HepRG hepatocyte line" (line 167). Do the authors mean HepaRG cells? HepaRG cells have been described as a hepatic progenitor cell line not a hepatocyte line. Please define the source how these cells, how they have been treated to differentiate them and how have been genetically modified.

The description of the method was revised to clarify the commercial source of these cells and manufacturer's protocols for finalizing the differentiation as an SOP.

13. Line 251: There is a reference to a submitted MS. Additional information should be provided about the anti-UIS4 antibody in the present study.

To Editor: *Nature Communications* allows for "manuscript in press" to be cited and our co-author, Sebastian Mikolajczak, provided a letter of support from.

The creation of recombinant monoclonal antibody is now a fairly routine technique. The anti-UIS4 recombinant monoclonal antibody used in this study was derived from a hybridoma expressing anti-UIS4 monoclonal antibody developed by methods previously described (Mikolajczak et al., *ibid*).

14. Line 343: typo "8-aminoquinolnes"

This has been amended.

REVIEWERS' COMMENTS:

Reviewer #1 (Remarks to the Author):

I am very pleased with the responses and I believe that the MS is robust and relevant enough to be published as it is now.

Reviewer #2 (Remarks to the Author):

This reviewer's critique was addressed satisfactorily with a few minor points missed (below). Also, a few very minor corrections suggested below:

Initial critique: Line 171-172 and Fig. 3a legend: Donor H is also susceptible to a single species (P.f.), but is not mentioned.

Response: The legend was revised accordingly.

Additional suggestion: Corrected in Figure legend but the text on line 172 is still missing mentioning donor H.

Initial critique: Line 309. Is word "immunization with" should be inserted after the word "from"? Same on line 312.

Response: This was revised accordingly.

Additional suggestion: Corrected on line 309 (now #317) but not in line 312 (now #321)

A few additional corrections suggested:

In supplementary Fig 7 legend: "Graph error bars represent means with s.d. from biological replicates (n = 2) and experimental replicates (n = 3)" word "error" should be removed.

In supplementary Fig 8 legend: "Graph error bars represent means with s.d. from..." word "error" should be removed. Also, four panels are not designated as a, b, c, or d.

In supplementary Fig 9 legend: "Graph error bars represent means with s.d. from ..." word "error" should be removed.

Reviewer #3 (Remarks to the Author):

The authors have largely addressed my concerns. This reviewer agrees that it is beyond the scope of the current study to define the mechanism by which hepatic functions appear to be better preserved in 384 well as opposed to larger well formats. While the authors have found a workable solution (screening of specific hepatocyte lots) they should be more explicit about the inherent shortcomings. In the future it may be desirable/necessary to screen/test compounds in hepatocytes of specific genotypes (e.g. certain cytochrome p450 combinations) and thus not being able to use any donor lot clearly creates a bottle neck.

The authors explicitly stated that "... [the] prodrug primaquine was found active only in lots with higher CYP2D6 activity (Supplementary Table 2, Supplementary Fig. 10)". Conceivably this might be an issue as some other compounds may be classified as false negatives. The authors suggested having addressed this issue by optimization of "dosing concentration and repetitions (maximum 10 μ M administered 3 times) ... to provide excessive opportunity for metabolically labile compounds to act on LSs." This reviewer agrees that this is a somewhat practical way of addressing this issue in the current platform but the general concern remains. Thus, it should be explicitly stated that false negatives cannot be ruled out.

RESPONSES TO REVIEWERS' COMMENTS:

Reviewer #1 (Remarks to the Author):

I am very pleased with the responses and I believe that the MS is robust and relevant enough to be published as it is now.

We thank the reviewer for their comments.

Reviewer #2 (Remarks to the Author):

This reviewer's critique was addressed satisfactorily with a few minor points missed (below). Also, a few very minor corrections suggested below:

We thank the reviewer for their comments.

Initial critique: Line 171-172 and Fig. 3a legend: Donor H is also susceptible to a single species (P.f.), but is not mentioned.

Response: The legend was revised accordingly.

Additional suggestion: Corrected in Figure legend but the text on line 172 is still missing mentioning donor H.

This was revised accordingly.

Initial critique: Line 309. Is word “immunization with” should be inserted after the word “from”? Same on line 312.

Response: This was revised accordingly.

Additional suggestion: Corrected on line 309 (now #317) but not in line 312 (now #321)

This was revised accordingly.

Revised:

Samples G6 and G7 were derived by immunization with a SAPN assembled from an FMP014V monomer containing two copies of the PvCSP (VKS type 210) repeat region motif and the \$\alpha\$ -TSR domain of PfCSP

A few additional corrections suggested:

In supplementary Fig 7 legend: “Graph error bars represent means with s.d. from biological replicates (n = 2) and experimental replicates (n = 3)” word “error” should be removed.

This was revised accordingly.

In supplementary Fig 8 legend: “Graph error bars represent means with s.d. from...” word “error” should be removed. Also, four panels are not designated as a, b, c, or d.

This was revised accordingly.

In supplementary Fig 9 legend: “Graph error bars represent means with s.d. from ...” word “error” should be removed.

This was revised accordingly.

Reviewer #3 (Remarks to the Author):

The authors have largely addressed my concerns. This reviewer agrees that it is beyond the scope of the current study to define the mechanism by which hepatic functions appear to be better preserved in 384 well as opposed to larger well formats. While the authors have found a workable solution (screening of specific hepatocyte lots) they should be more explicit about the inherent shortcomings. In the future it may be desirable/necessary to screen/test compounds in hepatocytes of specific genotypes (e.g. certain cytochrome p450 combinations) and thus not being able to use any donor lot clearly creates a bottle neck. The authors explicitly stated that "... [the] prodrug primaquine was found active only in lots with higher CYP2D6 activity (Supplementary Table 2, Supplementary Fig. 10)". Conceivably this might be an issue as some other compounds may be classified as false negatives. The authors suggested having addressed this issue by optimization of "dosing concentration and repetitions (maximum 10 μ M administered 3 times) ... to provide excessive opportunity for metabolically labile compounds to act on LSs." This reviewer agrees that this is a somewhat practical way of addressing this issue in the current platform but the general concern remains. Thus, it should be explicitly stated that false negatives cannot be ruled out.

We thank the reviewer for their comments.

This was revised accordingly in the Discussion.

Revision:

“Furthermore, the MMV compounds and Calibr Bioactive Library screened were all metabolism-optimized compounds, thus reducing the likelihood of a false negative due to metabolism. Despite these protocol adaptations to circumvent hepatic metabolism, we cannot rule out that our screen may miss quickly metabolized compounds.”(720-724).